# SciTS: Scientific Time Series Understanding and Generation with LLMs

**Wen Wu**[1], **Ziyang Zhang**[1,2], **Liwei Liu**[1,3], **Xuenan Xu**[1], **Jimin Zhuang**[1,2], **Ke Fan**[1,2],
**Qitan Lv**[1,4], **Junlin Liu**[1], **Chen Zhang**[1,2], **Zheqi Yuan**[1,2], **Siyuan Hou**[1,2], **Tianyi Lin**[1,2],
**Kai Chen**[1], **Bowen Zhou**[1,2], **Chao Zhang**[1,2] *

[1]Shanghai Artificial Intelligence Laboratory, [2]Tsinghua University
[3]Harbin Institute of Technology, [4]University of Science and Technology of China
{wuwen, zhangchao}@pjlab.org.cn

## Abstract

The scientific reasoning ability of large language models (LLMs) has recently attracted significant attention. Time series, as a fundamental modality in scientific data, presents unique challenges that are often overlooked in current multimodal LLMs, which either encode numerical sequences as text or convert them into images. Such approaches may be insufficient for comprehensive scientific time series understanding and generation. Existing unified time series models typically specialise in either forecasting or analysis, and their effectiveness on non-periodic, heterogeneous scientific signals remains unclear. To address these gaps, we introduce **SciTS**, a benchmark spanning 12 scientific domains and 43 tasks, with over 50k+ instances, both univariate and multivariate signals ranging from $10^0$ to $10^7$ in length and up to 10 MHz in frequency. We benchmark 17 models, including text-only LLMs, multimodal LLMs, and unified time series models, and find that general-purpose LLMs exhibit stronger generalisability than specialised time series models, while representing time series as text or images limits their performance due to excessively long sequences and loss of numerical precision, respectively. We then introduce **TimeOmni**, a working example to explore insights into how LLMs can be extended to handle scientific time series while remaining compatible with general-purpose LLM training. This work fills a gap in both dedicated benchmarks and illustrative frameworks for scientific time series, paving the way for LLMs to understand and generate complex temporal scientific data.[1]

## 1 Introduction

Large language models (LLMs) have demonstrated remarkable capabilities in natural language understanding, reasoning, and generation, and their potential for scientific applications has recently attracted increasing attention (Yang et al., 2025a; Zhou et al., 2025; Bai et al., 2025a). Scientific data, however, is often multimodal, and time series represents one of the most fundamental and widely encountered modalities across disciplines such as physics, astronomy, biology, and engineering. Understanding and generating scientific time series is critical for analysing diverse signals, such as astronomical light curves, neural recordings, and bioacoustic vocalisations, as well as tasks such as stellar collision detection and forecasting physiological states. However, scientific time series remain largely under-explored for LLMs. Bridging this gap is pivotal to transforming LLMs from narrative assistants into powerful engines that can interrogate dynamical systems, drive scientific discovery, and guide experimental design.

Current multimodal LLMs typically handle time series by either directly encoding numerical sequences as text or converting signals into images (Wang et al., 2025c; Bai et al., 2025b; Zhou et al., 2025). Representing time series as text leads to excessively long sequences that LLMs struggle

---

*Corresponding author.

[1]SciTS benchmark is available at: https://huggingface.co/datasets/OpenTSLab/SciTS.
TimeOmni framework is available at: https://github.com/OpenTSLab/TimeOmni.

to process, while image-based representations sacrifice numerical precision. As a result, these approaches may fail to capture the rich temporal dynamics, long-range dependencies, and domain-specific patterns inherent in scientific time series, limiting the models' ability to perform both comprehensive analysis and accurate generation. On the other hand, models specifically designed for time series have traditionally been developed for individual tasks, domains, or datasets, limiting their applicability beyond narrow settings. Unifying time series modelling across diverse domains is non-trivial, since signals exhibit varying scales, frequencies, dimensions, lengths, and dynamic patterns. Existing unified time series models make progress toward cross-domain applicability but typically focus on specific task types, such as forecasting (Shi et al., 2025; Jin et al., 2024; Wang et al., 2025b; Woo et al., 2024) or analytical tasks (Xie et al., 2025), and their effectiveness on non-periodic, heterogeneous scientific signals remains unclear. Moreover, these models often rely on specialised architecture designs that are not readily incorporated into LLMs to enhance their ability to handle time series.

To address these gaps, we first introduce **SciTS**, a large-scale benchmark for scientific time series understanding and generation. SciTS comprises 54,023 instances spanning 43 domain-specific tasks across 12 scientific disciplines: Astronomy, Bioacoustics, Earth Science, Economics, Energy, Manufacturing, Mathematics, Meteorology, Neuroscience, Physiology, Radar, and Urbanism. The benchmark covers both univariate and multivariate signals, with lengths ranging from $10^0$ to $10^7$ and frequencies up to 10 MHz, and includes 7 major task types: anomaly detection, classification, multiple-choice question answering, event localisation, forecasting, imputation, and synthesis.

We benchmark 17 state-of-the-art models on SciTS, spanning text-only and multimodal LLMs (open- and closed-weight) as well as unified time-series models. SciTS is highly challenging: instances exceed current system limits, leading to context-length overruns and instruction-following failures. While LLMs generalise better to scientific domains than specialised time-series models, serialising temporal signals as text or rasterising them as images constrains performance. TimeOmni attains the top rank, underscoring the advantage of explicitly modelling temporal dynamics within an LLM and the promise of LLM-based solutions for complex scientific time series.

In addition, we introduce **TimeOmni** as a working example to explore the key ingredients that may be necessary for unified scientific time series understanding and generation. TimeOmni leverages the reasoning and knowledge of LLMs while explicitly modelling temporal dynamics. Given an input time-series signal and a task prompt, it generates textual responses or time-series outputs. In order to handle diverse signals with varying lengths, resolutions, and dimensions, TimeOmni employs multiple patch experts with a novel routing mechanism that automatically selects the most suitable patch expert. The framework is intended to be readily incorporated into general-purpose LLMs for joint training with other modalities and tasks.

**Summary of Contributions. 1)** We introduce SciTS, to our knowledge the most comprehensive benchmark to date for scientific time series, spanning a wide range of domains, tasks, and signal types. **2)** We conduct a large-scale evaluation on SciTS, revealing key challenges in generalising to scientific domains and efficiently handling scientific time series. **3)** We introduce TimeOmni to study what is required for LLMs to handle diverse time-series data, revealing insights into explicit and adaptive temporal modelling.

## 2 RELATED WORK

**Science Time Series Benchmarks.** Time series benchmarks have been developed for tasks such as forecasting, anomaly detection, and question answering (QA) across domains including finance, weather, and traffic (Cai et al., 2024; Chen et al., 2025b; Liu et al., 2024b; Kong et al., 2025). However, scientific domains such as astronomy and earth science have not been extensively studied. Current scientific benchmarks for LLMs, such as Scientists' First Exam (Zhou et al., 2025), ScienceQA (Lu et al., 2022), and CHARTQAPRO (Masry et al., 2025), focus on reasoning over text and images, resulting in information loss when temporal data is converted into visual formats. SciTS addresses this gap by providing a large-scale, multi-domain benchmark focused on scientific time series understanding and generation, comparison to existing benchmarks shown in Table 1.

**Representation Learning for Time Series.** Developing unified time series modelling remains an open challenge. Time series span diverse domains and exhibit varied scales, sampling rates, and

Table 1: Comparison of SciTS and existing time series related benchmarks.

| Dataset | Input Format | Tasks | Scientific | Domains | Data Source | Multivariate | Size |
|---|---|---|---|---|---|---|---|
| SFE | Image | 2 | √ | 5 | Real | / | 830 |
| TimeSeriesExam | TimeSeries | 5 | × | / | Synthetic | × | 700 |
| Time-MQA | TimeSeries | 3 | × | 12 | Real | × | 200,000 |
| Time-MMD | TimeSeries | 3 | × | 9 | Real | × | 17,113 |
| MTBench | TimeSeries | 3 | × | 2 | Real | × | 20,000 |
| SciTS | TimeSeries | 7 | √ | 12 | Real & Synthetic | √ | 54,023 |

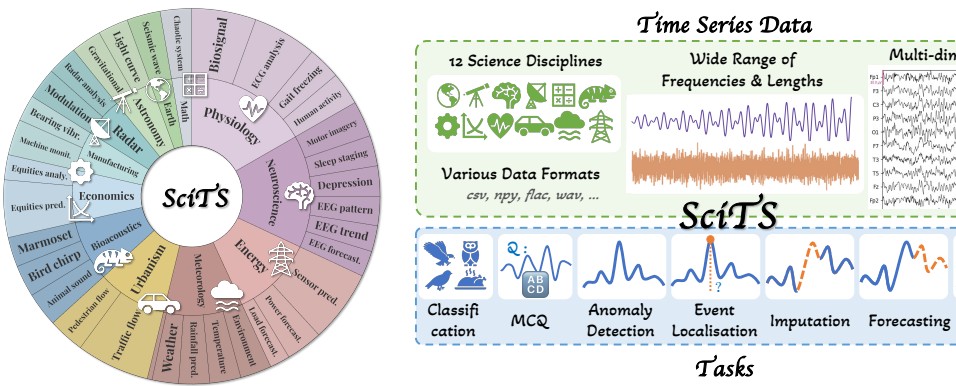

Figure 1: Twelve scientific disciplines included in SciTS.

Figure 2: Overview of SciTS, covering diverse frequencies, lengths, dimensions, and 7 task types.

dynamic patterns, complicating representation learning and posing challenges for models to adapt across tasks and domains. Transformer-based architectures form the backbone of unified time series models, including encoder-focused designs (Woo et al., 2024), decoder-based structures (Liu et al., 2025; Ansari et al., 2024; Wang et al., 2025b), and their extensions with mixture-of-experts (Xu et al., 2025a; Shi et al., 2025). LLMs have been recently explored for time series analysis, though their effectiveness remains debated: LLM-based forecasters often perform no better than strong non-LLM baselines, highlighting the difficulty of transferring linguistic pretraining to temporal prediction (Tan et al., 2024; Zhou & Yu, 2025). To bridge this gap, multimodal approaches treat time series as an additional modality (Xie et al., 2025), and alignment-based methods adapt temporal signals into language-compatible forms (Hu et al., 2025; Zhang et al., 2025; Jin et al., 2024). Despite these advances, existing approaches largely focus on either understanding or generation, rather than a unified treatment of both. UniTS (Gao et al., 2024) makes a step forward by integrating QA and forecasting tasks, while it relies on a separate architectural design, not compatible with general-purpose LLM training. Moreover, scientific signals, which are often heterogeneous and non-periodic, remain unexplored. In this work, we propose TimeOmni, a framework that unifies 7 types of understanding and generation tasks, and can be readily integrated into LLMs for joint training with other modalities and tasks. Extended discussion on the related work is provided in Appendix B.

## 3 SCITS DATASET AND TASKS

The SciTS benchmark consists of 54,023 samples spanning 43 tasks across 12 disciplines: Astronomy, Bioacoustics, Earth Science, Economics, Energy, Manufacturing, Mathematics, Meteorology, Neuroscience, Physiology, Radar, and Urbanism, as shown in Fig. 1. Time series from different domains exhibit widely varying characteristics: frequencies range from once per day in light curve observations to kilohertz for bird chirps in bioacoustics, and even megahertz in radar communications; lengths vary from fewer than 10 points for gait-freezing measurements to millions of points for animal sounds; moreover, SciTS includes multivariate signals, with the number of channels reaching up to 58 for electroencephalogram (EEG) recordings in neuroscience. Detailed dataset statistics are presented in Table 2, and more details can be found in Appendix D.2.

## 3.1 DATA COLLECTION

SciTS data was collected from a combination of open-source datasets, scientific domain websites, and numerical simulation methods widely used for relevant domains. Tasks were designed by leveraging the meta-information accompanying the data, which guided how inputs and target answers were defined. These tasks can be broadly grouped into two categories: *understanding*, where models answer questions given time series signals, and *generation*, where models produce time series conditioned on questions or question–signal pairs. The target answers in all cases were derived from the original meta-information or time series themselves, ensuring consistency with the underlying scientific data. All tasks were unified under a prompt-based format to ensure compatibility with LLM-style training and evaluation. Finally, systematic quality checks were conducted to verify the consistency, correctness, and scientific validity of the data.

## 3.2 DOMAINS

SciTS covers a wide range of scientific domains, each with distinct time series characteristics and tasks. In **Astronomy**, tasks involve detecting collapse or collision events and predicting merger times based on gravitational wave data, and classifying celestial objects based on light curve variability. In **Earth science**, tasks involve detecting earthquake events from seismic wave data and, if an event occurs, predicting the arrival times of P-waves and S-waves. In **Bioacoustics**, tasks involve classifying the species of birds or other animals based on their vocalizations, and identifying the type of marmoset call from recorded sounds. In **Meteorology**, tasks involve abnormal environmental event detection from parameters such as temperature, humidity, wind speed, and soil moisture, and precipitation forecasting based on historical measurements. **Economics** data consist of equity price series combined with news for trend prediction and causal analysis. In **Neuroscience**, tasks involve detecting neurological or psychiatric conditions from electroencephalogram (EEG), classifying EEG patterns, predicting or

Table 2: Statistics of SciTS

| Statistics | Number |
|---|---|
| # Discipline | 12 |
| # Domain tasks | 43 |
| # Instances | 54,023 |
| # Understanding tasks | 32,149 |
| # Classification | 18,364 |
| # MCQ | 716 |
| # Anomaly detection | 13,069 |
| # Generation tasks | 21,874 |
| # Forecasting | 15,593 |
| # Imputation | 2,014 |
| # Event localisation | 1,567 |
| # Synthesis | 2,700 |
| Numerical Value Range | $10^{-7} \sim 10^{6}$ |
| Frequency Range (Hz) | $10^{-5} \sim 10^{7}$ |
| Dimension Range | $1 \sim 58$ |
| Length Range | $10^{0} \sim 10^{7}$ |

imputing future EEG signals, identifying motor imagery, and sleep staging. In **Energy**, tasks involve short- and long-term prediction of power system behaviour based on half-hourly energy consumption readings, minute-by-minute wind power production, and hourly transformer sensor measurements. In **Physiology**, tasks involve analysing electrocardiogram (ECG) recordings to assess health status or detect abnormalities, recognising gait freezing from wearable accelerometer data, performing general activity recognition from multi-axis accelerometer measurements, and predicting or imputing physiological signals such as slow cortical potentials, heart rate, arterial or venous pressures, and atrial fibrillation events. In **Urbanism**, tasks involve predicting pedestrian activity from multi-sensor counts, forecasting traffic flow using historical measurements and relevant news, and detecting anomalies in traffic system data, including sudden surges or sensor malfunctions. In **Manufacturing**, tasks involve analysing vibration signals from industrial bearings to determine the size and location of damaged components, as well as detecting anomalies or malfunctions from operational recordings. In **Radar**, tasks involve classifying radar activities based on coding schemes and categorising operational modes and communication modulations. In **Mathematics**, tasks involve forecasting the evolution of multivariate chaotic systems based on given input.

## 3.3 TYPE OF TASKS

SciTS covers 7 major types of tasks, described as follows. *Understanding* tasks include **anomaly detection**, where models determine whether abnormal events occur in a given time series; **classification**, where models determine which category (or categories) the input series belongs to from a given set; and **multiple-choice question answering (MCQ)**, where models are given a time series and a

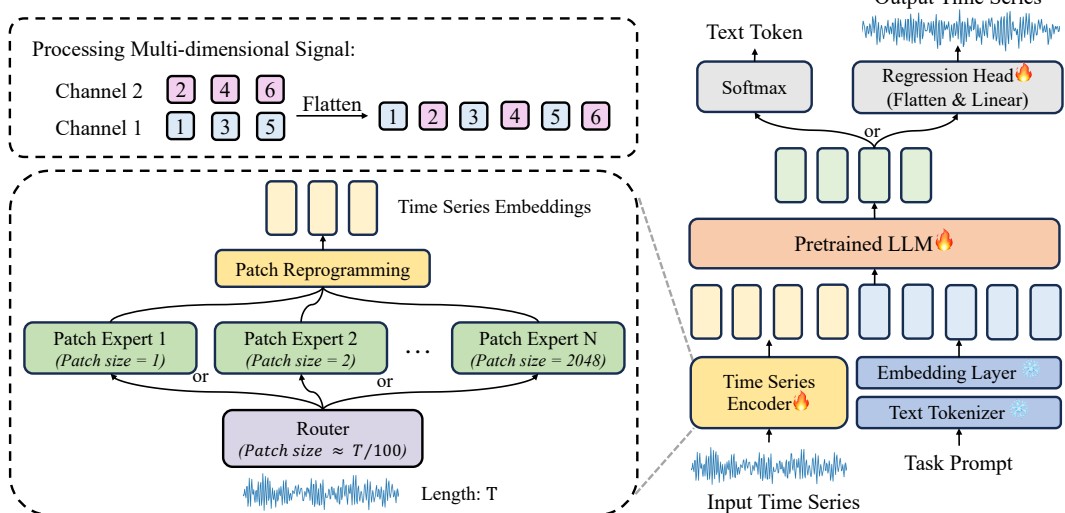

Figure 3: The overall architecture of TimeOmni. An input time series is processed by the Time Series Encoder, where a Router selects an appropriate Patch Expert, and embeddings are refined via Patch Reprogramming. In parallel, the Task Prompt is processed through the Text tokeniser and the Embedding Layer of LLM. The time series and prompt embeddings are concatenated and fed into the Pretrained LLM. Two output heads used: (i) understanding tasks produce text via a softmax layer, and (ii) generation tasks generate time series through a flattening and linear transformation. The way to process multi-dimensional signals are shown in top left.

question with four options (A–D) and must select the correct answer. *Generation* tasks involve **event localisation**, identifying the timestamps of specific events within a time series; **forecasting**, predicting future values based on observed data; **imputation**, completing series with missing values; and **synthesis**, generating time series from textual descriptions. The inputs can be context-dependent, including auxiliary text such as background information or news to guide the output, or context-independent, where only the time series is provided. Generated sequences range from short-term predictions of a few steps to long-term forecasts of up to 720 steps. Further details on signal types, frequencies, and lengths for each task are provided in Appendix D.

## 4 TIMEOMNI: A WORKING EXAMPLE

To explore the key ingredients that may be necessary for extending current LLMs to handle scientific time series, we develop TimeOmni as an illustrative framework. The structure of TimeOmni is shown in Fig. 3, consisting of a time series encoder, a LLM backbone, and output heads for diverse signals. Given an input time series signal $\mathbf{X} \in \mathbb{R}^{T' \times N}$, it is first flattened along the time dimension to obtain $\mathbf{X}' \in \mathbb{R}^{NT' \times 1}$. The encoder automatically selects a patch expert based on $NT'$, producing $\mathbf{X}_{\text{enc}} \in \mathbb{R}^{T_{\text{enc}} \times D_{\text{llm}}}$, where $D_{\text{llm}}$ is the LLM hidden dimension and $T_{\text{enc}}$ is typically 100 to 200. The task prompt is tokenised and embedded into $\mathbf{P} \in \mathbb{R}^{L \times D_{\text{llm}}}$, which is concatenated with $\mathbf{X}_{\text{enc}}$ as input to the LLM backbone. Output embeddings from the LLM are processed according to task type: a softmax layer to generate discrete text tokens for understanding tasks and a regression head to output a time series for generation tasks.

### 4.1 TIME SERIES ENCODER

As shown in Fig. 3, , time series is explicitly modelled by an encoder, which consists of three main components: a router, a patch expert family, and a patch reprogramming module (Jin et al., 2024). For a flattened input of length $T = NT'$, the Router selects a patch size $D_{\text{patch}}$ to ensure that the output sequence length after the Patch Expert layer falls between 100 and 200, *i.e.*, $T/200 < D_{\text{patch}} < T/100$. The Patch Expert first patchifies the signal, reshaping $\mathbf{X}'$ from $\mathbb{R}^{T \times 1}$

to $\mathbb{R}^{\lceil T/D_{\text{patch}}\rceil \times D_{\text{patch}}}$. A 1-dimensional (-dim) convolution then maps it to $\mathbf{X}_{\text{patch}} \in \mathbb{R}^{\lceil T/D_{\text{patch}}\rceil \times D_{\text{enc}}}$, where $D_{\text{enc}}$ is consistent across the Patch Expert family. The Patch Reprogramming module re-represents the time series using the LLM's vocabulary embeddings $\mathbf{E} \in \mathbb{R}^{\text{vocab\_size} \times D_{\text{llm}}}$. $\mathbf{E}$ is first projected through a linear layer to $\mathbb{R}^{1000 \times D_{\text{llm}}}$, then $\mathbf{X}_{\text{patch}}$ attends to $\mathbf{E}$ via multi-head cross-attention, with $\mathbf{X}_{\text{patch}}$ as query and $\mathbf{E}$ as key and value. A final linear projection produces the encoder output $\mathbf{X}_{\text{enc}}$.

## 4.2 LLM Input and Output

For understanding tasks, we use a *Prompt-as-suffix* strategy, where $\mathbf{X}_{\text{enc}}$ is followed by $\mathbf{P}$. For generation tasks, a *Prompt-as-prefix* approach is adopted, placing $\mathbf{P}$ before $\mathbf{X}_{\text{enc}}$. The concatenated vector is fed into the LLM. For understanding tasks, output embeddings of LLMs pass through a softmax layer to produce discrete text tokens, following the standard LLM generation process. For generation tasks, the output embeddings are flattened and mapped via a linear layer to the required length of the output time series. To handle the wide range of output lengths in the benchmark, we predefine a set of regression heads and the model selects the one with the closest length, truncation applied if necessary.

## 5 Experimental Setup

**General Settings.** We conduct a comprehensive evaluation of various types of models on SciTS. This includes **(i) Text LLMs** with closed and open weights: GPT-4.1-mini, Gemini-2.5-Flash, DeepSeek-V3 (Liu et al., 2024a), Llama3-8B (Grattafiori et al., 2024), Qwen3-4B (Yang et al., 2025a), Qwen3-8B (Yang et al., 2025a); **(ii) Multimodal LLMs** with closed and open weights: GPT-5-mini, Gemini-2.5-Flash, InternVL3.5-4B (Wang et al., 2025c), InternVL3.5-8B (Wang et al., 2025c), Qwen2.5-VL-7B (Bai et al., 2025b); **(iii) Unified time series models**: Moirai-Large (Woo et al., 2024), TimeMoE-Large (Shi et al., 2025), Chronos-bolt-Base (Ansari et al., 2024), ChaTS (Xie et al., 2025), UniTS (Gao et al., 2024). For text LLMs, time series are processed as sequences of digits in text form. For multimodal LLMs, time series are converted into an image. Compared to LLMs, unified time-series models are specifically tailored to time-series data, often trained across multiple domains and datasets, but typically specialise in particular applications such as forecasting. All models were assessed in a zero-shot setting. Details can be found in Appendix E. **TimeOmni** is initialised with pretrained weights of Qwen3-8B (Yang et al., 2025a) and finetuned using DoRA (Liu et al., 2024c). Implementation details can be found in Appendix C.

**Metrics.** We present accuracy and F1 for all understanding tasks. For generation tasks, we report mean absolute error (MAE), mean absolute percentage error (MAPE), success rate, and success-rate-weighted MAPE (swMAPE). MAPE measures the relative prediction error between predicted $\hat{y}_i$ and actual values $y_i$, expressed as a percentage of the actual values: $\text{MAPE} = \frac{100\%}{n} \sum_{i=1}^{n} \left| \frac{y_i - \hat{y}_i}{y_i} \right|$. Models can sometimes fail to process a given input instance, particularly for generation tasks, *e.g.*, when the input exceeds the model's maximum context window or when the model fails to follow instructions. To account for such cases, we report the success rate for each generation task, defined as the proportion of successfully processed samples relative to the total number of samples. We then define swMAPE as MAPE divided by the success rate, providing an overall performance measure for generation tasks that penalises failures. Note that all MAE and MAPE values are computed on the original data scale, not on normalised (*e.g.*, z-scored) inputs.

## 6 Results and Discussion

We assess 17 models mentioned in Section 5 on SciTS across 43 domain-specific tasks spanning 12 disciplines, 7 task types, and a wide range of frequencies, lengths, and multivariate signals.

## 6.1 Task Coverage and Success Rate

We first examine the ability of the assessed models to handle the diverse task set. As shown in Fig. 4a, closed-source LLMs, such as GPT-mini and Gemini2.5-Flash, can perform most tasks using either text or image inputs. In contrast, open-source text-only LLMs (*e.g.*, Qwen, LLaMA) fail on

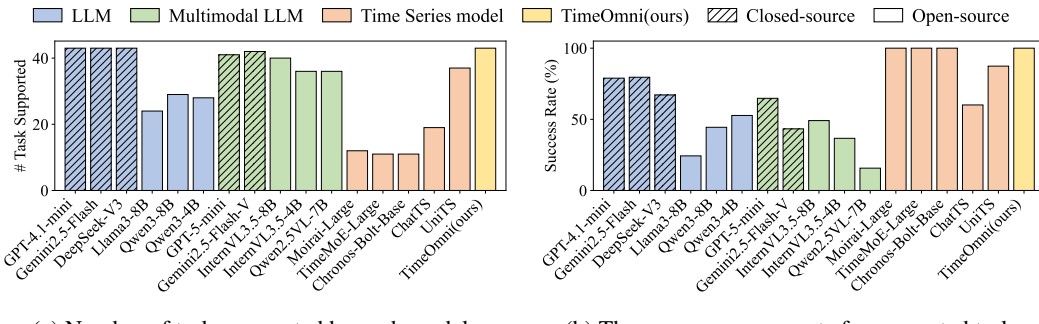

(a) Number of tasks supported by each model.

(b) The average success rate for supported task.

Figure 4: Task coverage (a) and average success rate (b) for models on SciTS. (a): fraction of supported domain tasks for each model. (b): average instance-level success rate for supported tasks.

approximately 10% of tasks. This is primarily due to three factors: (i) input sequences exceeding the maximum context length, which affects signals such as bioacoustic vocalisations and gravitational waves; (ii) failures to process multivariate signals such as EEG and generate multivariate sequences such as chaotic system parameters; (iii) failures to follow output instructions to generate sequences of the required length. This limitation is partially mitigated when time series are visualised as images for multimodal LLMs that accept image inputs, shown by higher task coverage.

While closed-source LLMs support most task types, they still face instance-level limitations: handling a task does not guarantee successful processing of every individual case. To quantify this, we report success rates in Fig. 4b, showing that all LLM-based methods exhibit relatively low success rates at the instance level, with closed-source models generally outperforming open-source ones.

On the other hand, unified time series models typically focus on a certain type of task, resulting in low overall task coverage (Fig. 4a). For example, Moirai and TimeMoE only support forecasting and cannot handle imputation tasks or tasks requiring text input. Despite this limitation, they achieve high success rates on the tasks they do support, demonstrating the effectiveness of specialised models. The illustrated TimeOmni framework, in comparison, successfully handles all tasks and instances, achieving both full coverage and full success across the benchmark.

## 6.2 UNDERSTANDING TASKS

Table 3 compares text-only LLMs, multimodal LLMs, and time series models on understanding tasks across different scientific disciplines, with results reported as the average F1 score across tasks in each discipline. Overall, models achieve limited performance on scientific domains such as astronomy, neuroscience, and physiology, possibly due to these scientific domains being largely absent from the training data of general-purpose models. Performance is particularly low for bioacoustics and radar, with F1 scores below 10%, highlighting the challenging nature of the SciTS benchmark. In contrast, nearly all models achieve strong performance on economics (except LLaMA3), as this domain is closer to general knowledge and thus falls within the scope of general-purpose models. It is worth noting that for the urbanism domain, the results in Table 3 are primarily from Task URU04[2], an anomaly detection task with relatively few instances. Most models effectively perform random guessing, and the highest F1 score (bolded) largely reflects a bias toward a single class. In fact, GPT-4.1-mini and Qwen-2.5-VL achieve higher accuracy with detailed results provided in Appendix F.

Overall, multimodal LLMs that take images as input outperform text-only LLMs, possibly because tasks requiring high-level understanding do not strongly depend on precise numerical values, and representing time series as text can produce extremely long inputs that are difficult for LLMs to process. Converting time series into images compresses these long sequences more effectively. We also found models specialised for time series do not necessarily surpass general-purpose LLMs across many domains, which indicates that when exposed to data domains unseen during training, LLMs demonstrate stronger generalisation capabilities. Across most domains, the illustrated TimeOmni framework achieves the overall best performance, with Gemini2.5-Flash ranking second.

---

[2]The mapping of task IDs to their corresponding tasks is provided in Table 6.

Table 3: Performance on understanding tasks of SciTS. Values show the average F1 (%) across tasks within each discipline (accuracy used for MCQ tasks). If a model fails any task in a discipline, the fraction completed (completed/total) is reported. Average ranking (AvgRk) computed by ranking models for each task and averaging across all understanding tasks.

| | Astron. | Bioacous. | EarthSci. | Econ. | Meteoro. | Manuf. | Neuro. | Physio. | Radar | Urban. | AvgRk |
|---|---|---|---|---|---|---|---|---|---|---|---|
| *Large Language Models (Text Input)* | | | | | | | | | | | |
| GPT-4.1-mini | 41.4 | 6.7 | 67.0 | 90.4 | 45.3 | 31.7 | 13.5 | 26.8 | 17.6 | 64.4 | 6.1 |
| Gemini2.5-Flash | 40.2 | 10.3 | 67.6 | 87.8 | 51.8 | 28.8 | 12.7 | 31.8 | 17.2 | 64.6 | 5.5 |
| DeepSeek-V3 | 6.7 | 2.2 | 40.2 | 91.1 | 49.9 | 20.8 | 9.6 | 27.0 | 11.8 | 64.7 | 7.7 |
| Llama3-8B | (1/2) | (0/3) | (0/1) | 51.2 | 49.1 | (0/3) | (0/4) | 32.6 | 9.2 | **67.4** | 12.5 |
| Qwen3-4B | (1/2) | (0/3) | 46.5 | 77.9 | 40.9 | (0/3) | (1/4) | 30.0 | 8.3 | 62.8 | 12.4 |
| Qwen3-8B | (1/2) | (0/3) | 47.2 | 83.5 | 54.3 | (0/3) | (1/4) | 33.4 | 6.5 | 66.2 | 11.8 |
| *Multimodal Large Language Models (Text + Image Input)* | | | | | | | | | | | |
| GPT-5-mini | 42.3 | 10.7 | 67.6 | 83.8 | 45.3 | 38.4 | 13.9 | 25.0 | 16.5 | 64.8 | 6.0 |
| Gemini2.5-Flash | 38.4 | 7.7 | 72.5 | 82.9 | 52.3 | 39.4 | 29.8 | 27.3 | 21.4 | 67.0 | 5.8 |
| InternVL3.5-4B | 39.7 | 3.8 | 73.6 | 86.1 | 36.4 | 11.7 | 15.4 | 28.2 | 4.8 | **67.4** | 8.6 |
| InternVL3.5-8B | 21.3 | 4.7 | 81.0 | 85.8 | 54.0 | 19.6 | 23.9 | 30.0 | 3.1 | **67.4** | 6.9 |
| Qwen2.5-VL-7B | 7.1 | 9.3 | 80.6 | 81.9 | 28.1 | 6.4 | 6.0 | 16.7 | 5.9 | 58.5 | 9.5 |
| *Time Series Models* | | | | | | | | | | | |
| ChaTS | 11.3 | (0/3) | 64.8 | 79.2 | 51.2 | (2/3) | 22.7 | 30.9 | 13.9 | 65.4 | 9.2 |
| UniTS | 38.2 | 8.1 | 0.0 | 27.1 | 9.8 | 48.5 | 25.9 | 22.9 | 10.6 | **67.4** | 7.9 |
| TimeOmni (Ours) | **73.2** | **58.1** | **82.5** | **96.4** | **61.3** | **82.0** | **60.1** | **45.9** | **68.9** | 64.8 | **1.9** |

Table 4: Performance on generation tasks of SciTS. Values show the average success rate weighted MAPE (swMAPE) across tasks within each discipline. If a model fails any task in a discipline, the fraction completed (completed/total) is reported. Average ranking (AvgRk) computed by ranking models for each task and averaging across all generation tasks.

| | Astron. | EarthSci. | Meteoro. | Econ. | Neuro. | Energy | Physio. | Urban. | Math | AvgRk |
|---|---|---|---|---|---|---|---|---|---|---|
| *Large Language Models (Text Input)* | | | | | | | | | | |
| GPT-4.1-mini | 100.9 | 65.0 | 85.0 | 112.2 | 61.4 | 2.0e3 | 610.6 | 670.0 | 1.2e3 | 6.7 |
| Gemini2.5-Flash | 116.6 | 63.0 | 107.5 | 4.5 | **38.7** | 307.6 | **60.5** | **391.4** | 477.5 | 4.6 |
| DeepSeek-V3 | 584.4 | 90.1 | 150.3 | 8.4 | 77.0 | 101.0 | 114.4 | 1.0e3 | 594.5 | 6.3 |
| Llama3-8B | (0/1) | (0/1) | (0/1) | (0/2) | 2.8e3 | 199.6 | 1.1e3 | 1.7e4 | (0/1) | 12.9 |
| Qwen3-4B | (0/1) | 7.9e4 | (0/1) | (1/2) | 153.9 | 110.0 | 293.7 | 1.4e3 | (0/1) | 9.9 |
| Qwen3-8B | (0/1) | 2.9e4 | 1.6e4 | (1/2) | 242.4 | 148.2 | 280.8 | 4.8e3 | (0/1) | 10.2 |
| *Multimodal Large Language Models (Text + Image Input)* | | | | | | | | | | |
| GPT-5-mini | 55.4 | 25.4 | 72.6 | (1/2) | 1.1e3 | (4/5) | 230.9 | 558.0 | 1.5e3 | 8.1 |
| Gemini2.5-Flash | 108.7 | 24.6 | 142.5 | 10.8 | 330.5 | (4/5) | 4.9e3 | 2.1e3 | 521.0 | 9.8 |
| InternVL3.5-4B | 80.5 | 275.6 | 1.7e3 | (0/2) | 809.2 | (4/5) | 4.2e3 | (3/4) | (0/1) | 13.4 |
| InternVL3.5-8B | 167.8 | 219.6 | 137.7 | (1/2) | 472.3 | (4/5) | 715.7 | 1.3e3 | (0/1) | 10.8 |
| Qwen2.5-VL-7B | 445.3 | 53.7 | 1.8e4 | (1/2) | 2.0e4 | (1/5) | 5.6e3 | (3/4) | (0/1) | 14.4 |
| *Time Series Models* | | | | | | | | | | |
| Moirai-Large | (0/1) | (0/1) | 51.7 | **1.8** | (1/2) | (3/5) | (1/2) | (3/4) | **360.1** | 8.3 |
| TimeMoE-Large | (0/1) | (0/1) | 39.0 | (1/2) | (1/2) | (3/5) | (1/2) | (3/4) | 1.2e3 | 9.1 |
| Chronos-bolt-Base | (0/1) | (0/1) | 41.5 | (1/2) | (1/2) | (3/5) | (1/2) | (3/4) | 440.4 | 8.7 |
| UniTS | 3.3e6 | (0/1) | 42.0 | (1/2) | 147.3 | (4/5) | 216.3 | (3/4) | (0/1) | 9.8 |
| TimeOmni(Ours) | **2.8** | **2.2** | **37.5** | 5.3 | 46.6 | **66.4** | 91.7 | 402.7 | 656.5 | **4.1** |

## 6.3 GENERATION TASKS

Table 4 compares various models on generation tasks. Overall, generation tasks prove more challenging, as reflected by the large swMAPE values reported in the table. Unlike understanding tasks, representing time series as text generally outperforms image-based inputs for generation tasks where precise numerical values are critical for accurate generation, as expected. Gemini2.5 performs best in neuroscience, physiology, and urbanism, where the output time series are relatively short (typ-

ically fewer than 20 steps[3]). For long-term forecasting tasks, such as Task ENG02 with output lengths up to 720, all LLMs struggle to generate sequences of the required length, with success rates commonly below 10% (Table 12). Closed-source LLMs continue to outperform open-source counterparts across most domains. Many values for unified forecasting models are missing in the table, as they cannot handle imputation or event localisation tasks. However, they achieve superior performance on supported forecasting tasks, with Moirai giving the overall lowest average swMAPE in economics and mathematics. The proposed TimeOmni model achieves the highest overall ranking, closely followed by Gemini2.5-Flash using text input. Detailed results for each task are provided in Appendix F.

## 6.4 MODEL RANKINGS ACROSS DISCIPLINES

Fig. 5 shows the average rankings of 17 tested models across 12 disciplines. We rank models for each of the 43 tasks and average the ranking over tasks within each discipline. The benchmarking results reveal several clear patterns. General-purpose LLMs such as GPT-4.1-mini, Gemini2.5-Flash, DeepSeek-V3, and GPT-5-mini deliver strong all-around results, usually falling within the top 5 to 8 ranks. Multimodal LLMs like InternVL and QwenVL occupy a middle ground, showing competitive performance in sensory-heavy fields such as bioacoustics, earth science, and radar, but weaker results in other areas. Specialised time-series models, including Moirai, TimeMoE, Chronos, and ChatTS, struggle with generalisation. They often collapse to the bottom of the ranking across most domains due to limitations of supported task types, but occasionally excel in specific cases, such as Moirai and Chronos in mathematics and ChatTS in earth science. Discipline-specific patterns also emerge: models consistently perform well across language-heavy domains such as Economics, while dedicated time-series models show strengths in more structured or periodic domains like Energy and Meteorology, but degrade sharply elsewhere. As an illustrative framework, TimeOmni achieves consistently top performance across nearly all disciplines, with slightly lower in mathematics and urbanism.

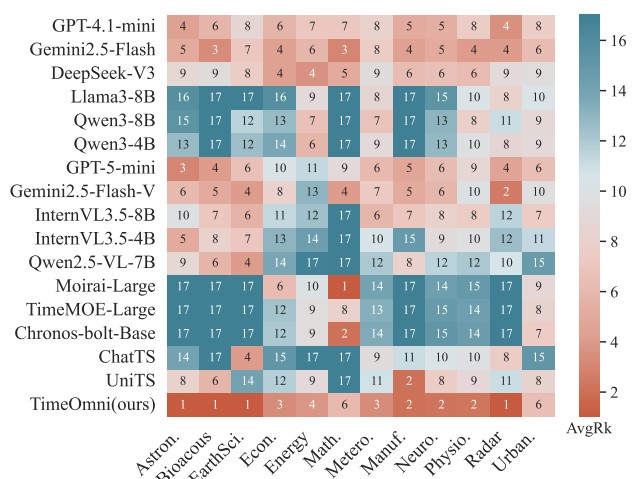

Figure 5: Average model rankings across disciplines, computed as the mean rank over all domain tasks within each discipline, rounded for visualisation purposes.

These results suggest two key insights. First, time series should be treated as a distinct modality rather than being directly converted to text or images, as such transformations lead to information loss and limit model performance. Second, general-purpose LLMs demonstrate better generalisation to unseen domains compared with specialised time-series models, highlighting the benefit of combining LLMs with dedicated time series processing to effectively handle diverse scientific data. TimeOmni provides an illustrative step that incorporates time series into LLMs while remaining compatible with general LLM training.

## 6.5 DESIGN ANALYSIS: WHAT IS NEEDED FOR SCIENTIFIC TIME SERIES MODELLING?

To better understand what is essential for modelling scientific time series, we first conducted ablation experiments on the reprogramming module and patch expert routing mechanism of TimeOmni. Due to space constraints, we summarise main findings here and defer complete experimental results to Appendix G. Replacing the reprogramming module with a simple MLP consistently degrades performance, suggesting that explicit reprogramming is important for aligning time series represen-

---

[3]Detailed statistics of each task can be found in Table 7.

tations with the LLM and improving robustness across tasks. Replacing patch expert routing with a fixed patch size faces a fundamental dilemma: small patch sizes become infeasible for very long sequences due to memory constraints and difficulty of modelling extremely long-range dependencies, while large patch sizes collapse short sequences into a single patch. Choosing an intermediate patch size leads to clear performance drops on sequences with extreme temporal scales, highlighting the need for scale-adaptive patching in scientific time series.

To complement the above analysis, we further investigate whether existing models can be directly adapted to scientific time series through finetuning. We additionally finetuned one image–text LLM (Qwen2.5VL-7B) and one time-series foundation model (TimeMoE) using the same training data as TimeOmni. Finetuning Qwen2.5VL improves instruction-following ability on some tasks, but the gains are limited possibly because representing time series as images compromises numerical precision. We also observed instability during finetuning and difficulty in maintaining balanced performance across tasks, potentially reflecting the limited capacity of image-based representations for time-series data. We then finetune TimeMoE, a forecasting-oriented model. While performance improves on some forecasting tasks, finetuning does not enable it to handle tasks it could not previously address, such as imputation or QA. In sum, results provide evidence that challenges arises from architectural limitations and cannot be overcome through finetuning alone. Full results and discussion are provided in Appendix H.

# 7 CONCLUSIONS

We introduce SciTS, a benchmark for scientific time series understanding and generation, spanning 12 disciplines, 7 task types, 43 domain-specific tasks, and over 50k instances, with a wide range of frequencies, lengths, and multivariate dimensions. We benchmark 17 models on SciTS, including text-only LLMs, multimodal LLMs, and unified time series models. Two key insights emerge. First, LLMs exhibit stronger adaptability and generalisation compared with models specialised for time series. Second, while general-purpose LLMs generalise better to unseen domains, pairing them with dedicated temporal processing is crucial to handle complex scientific dynamics. Serialising temporal signals as text or rasterising them as images incurs information loss and hinders efficiency and numerical fidelity. To this end, we provide TimeOmni, a demonstrated framework that integrates time series into LLMs while remaining compatible with general-purpose LLM training. By providing a comprehensive benchmark and a preliminary approach, this work takes a step toward LLMs that can faithfully interrogate dynamical systems, enable discovery, and guide experimental design on real-world scientific temporal data.

## ACKNOWLEDGMENTS

Supported by Shanghai Artificial Intelligence Laboratory. We thank the PrismaX Team[4] for their support and valuable advice in building SciTS.

---

[4] https://huggingface.co/PrismaX

## LIMITATIONS AND ETHICS STATEMENT

Due to resource constraints, we evaluate all baselines in a zero-shot setting without finetuning. More explanation of the evaluation setting can be found in Appendix H. In addition, our preliminary small-scale experiments indicate that enabling the "thinking" mode in closed-source LLMs does not improve performance on time-series tasks. Since it also incurs substantial computational cost, we disable this option for model assessments.

This work focuses on benchmarking and modelling methods for scientific time series. No personally identifiable or sensitive human data are included. As with most research in machine learning, new modelling techniques could potentially be misused by malicious actors; however, we do not believe that TimeOmni poses risks beyond those associated with existing methods.

## REPRODUCIBILITY STATEMENT

Additional statistics and details of SciTS are provided in Appendix D, with case examples in Appendix I. A detailed description of the assessed models is given in Appendix E, and implementation details of TimeOmni are presented in Appendix C. The SciTS benchmark is available at `https://huggingface.co/datasets/OpenTSLab/SciTS`. The TimeOmni framework is available at `https://github.com/OpenTSLab/TimeOmni`.

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

## A  The Use of Large Language Models (LLMs)

LLMs were employed solely to polish the writing and correct grammatical errors.

## B  Extended Related Work

### B.1  Time Series Models

**Traditional Time Series Models.** Traditional task-specific models are typically trained in a supervised manner on individual datasets, requiring carefully designed architectures and specialized output modules tailored to specific tasks. For classification tasks such as anomaly detection, regime identification, or event recognition, models commonly employ a final classification head whose structure depends directly on the number of predefined classes in the target dataset (Lu et al., 2023; Liu et al., 2023; Ding et al., 2023). This design choice limits their generalization ability, as the model must be reconfigured and retrained whenever applied to a new classification scenario with different class cardinality. Similarly, time series forecasting and imputation models are often developed and optimized in isolation. Architectures such as iTransformer (Liu et al., 2024d), which reformulate Transformer-based modelling through inverted attention for improved long-term prediction, and TimeMixer (Wang et al., 2024a), which introduces hybrid temporal projections for efficient multi-scale forecasting, are typically trained on narrowly defined forecasting benchmarks. Likewise, imputation methods like Kim et al. (2023), Tactis-2 (Ashok et al., 2024) are usually evaluated only on missing-data reconstruction tasks using domain-specific datasets. As a result, these models lack cross-task compatibility and fail to develop a unified understanding of temporal patterns across diverse modalities and applications. This fragmentation leads to inefficiencies in deployment, scalability, and maintenance, especially when multiple time series tasks need to be performed simultaneously in real-world systems.

**Unified Time Series Models.** Moirai (Woo et al., 2024) propose the Masked Encoder-based Universal Forecasting Transformer architecture, capable of processing various time series signals. Models building on this, Moirai-MoE (Xu et al., 2025a) and Time-MoE (Shi et al., 2025) integrate Mixture of Experts (MoE) into the Transformer framework, enabling specialized subnetworks to capture heterogeneous temporal patterns through dynamic routing. Concurrently, TimeMixer++ (Wang et al., 2025b) advances the paradigm by capturing task-adaptive temporal patterns through a multi-scale decomposition strategy, simultaneously modelling intricate dependencies in both the time and frequency domains. This dual-domain analysis allows the model to extract features at varying temporal resolutions, enhancing its ability to generalize across tasks with differing periodicities and noise profiles. In contrast to architectural overhauls of large Transformer backbones, SimpleTM (Chen et al., 2025a) adopts a more principled and minimalist design philosophy. It integrates classical signal processing concepts—such as local averaging and trend-seasonal decomposition—directly into a lightweight neural architecture, complemented by a carefully modified attention mechanism. Rather than introducing complex modifications to a pre-existing large-scale model, SimpleTM incrementally incorporates domain-inspired components from scratch, achieving competitive performance with significantly reduced model complexity. Addressing the pervasive challenge of missing data in real-world time series, S4M (Peng et al., 2025) introduces a prototype-based memory mechanism that leverages rich historical patterns stored in a learnable prototype bank. By encoding both observed values and binary masks indicating missingness, the model learns robust and context-aware representations that are resilient to data incompleteness. The integration of explicit missingness indicators with structured memory enables more accurate imputation and forecasting.

**LLMs for Time Series Modelling.** Recent studies have critically examined the efficacy of large language models (LLMs) in time series forecasting and reasoning. Tan et al. (2024) argue that LLM-based time series forecasters perform on par with—or even worse than—basic LLM-free ablations, suggesting that the pretrained knowledge in LLMs may not be effectively leveraged for temporal prediction. Zhou & Yu (2025)'s comprehensive analysis of LLMs' understanding of time series using anomaly detection also suggests that while LLMs can identify simple anomalies, their capacity for numerical reasoning and deep temporal comprehension remains severely constrained. In response, several works propose multimodal or alignment-based frameworks to better interface time series data with LLMs. ChatTS (Xie et al., 2025) introduces a novel multimodal LLM architecture that treats time series as a first-class modality—akin to images in vision-language models, enabling

end-to-end understanding and reasoning over temporal data. Another line of work focuses on semantic and structural alignment. Hu et al. (2025) propose a paradigm that aligns time series data with linguistic representations within the familiar context space of LLMs, leveraging the models' inherent strengths in logical and structural reasoning to improve temporal comprehension. Building on this idea, TempoGPT (Zhang et al., 2025) quantizes temporal embeddings into discrete tokens and expands the LLM's embedding layer to jointly process temporal and textual inputs, achieving a unified representation space. Similarly, TIME-LLM (Jin et al., 2024) presents a framework to adapt LLMs for time series forecasting without modifying the pretrained backbone. It reprograms raw time series into text-like prototype representations, enriching them with contextual descriptions and task instructions in natural language, thereby aligning temporal data with the LLM's linguistic inductive biases.

### B.2 SCIENCE BENCHMARKS AND TIME SERIES BENCHMARKS

**Science Data Benchmarks.** Scientists' First Exam (SFE) (Zhou et al., 2025) comprises 830 expert-validated Visual Question Answering (VQA) pairs spanning five high-value scientific disciplines—astronomy, chemistry, earth science, life science, and materials science—and incorporating 17 distinct scientific data formats, such as spectra, phase diagrams, and remote sensing images. ScienceQA (Lu et al., 2022) focuses on reasoning capabilities over educational or general-purpose data visualizations. It comprises approximately $21,000$ multimodal (text, image) multiple-choice questions, which feature explanations in Chain-of-Thought (Wei et al., 2022) format to simulate multi-hop reasoning processes. ChartQA (Masry et al., 2022), along with its subsequent version CHARTQAPRO (Masry et al., 2025), concentrates on VQA for charts, necessitating visual and logical reasoning capabilities from the models. Notably, CHARTQAPRO expands data diversity by introducing $1,341$ charts from 99 different sources, covering complex formats like infographics and dashboards, while also integrating question types designed to be more challenging. However, an in-depth analysis of these image input benchmarks uncovers a "visualization gap. This circumvents the entire data processing and analysis stage.

**Time Series Benchmarks.** Existing time series benchmarks mainly provide contextual information for the data by integrating time series with text. TimeSeriesExam (Cai et al., 2024) contains 700 questions generated from 104 carefully curated templates and provides a set of configurable tasks designed to evaluate a model's understanding of core concepts such as causality, patterns, and noise. MTBench (Chen et al., 2025b) aligns financial news and weather reports with their corresponding stock price and temperature time series, which supports tasks such as news-driven question answering and trend analysis tasks. Time-MMD (Liu et al., 2024b) further extends by offering the first large-scale, multi-domain dataset, which covers 9 major sectors (such as agriculture, energy, and healthcare) and enables fine-grained alignment between numerical and textual sequences. Time-MQA introduces a unified, large-scale question answering framework, including approximately $200,000$ questions across more than 12 domains. However, synthetic data benchmarks (e.g., TimeSeriesExam) provide high controllability and interpretability, yet this approach may fall short of generalizing to real-world data. Existing real-world data benchmarks (e.g., Time-MQA) offer authenticity but may lack the systematic coverage of synthetic methods. Our SciTS achieves a balance of both: it is sourced from 11 scientific domains and 40 tasks, ensuring models are exposed to a broad spectrum of real-world phenomena and also apply synthetic data for a systematic consideration.

## C IMPLEMENTATION DETAILS OF TIMEOMNI

We initialise TimeOmni with the pretrained weights of Qwen3-8B (Yang et al., 2025a) and finetune it using DoRA (Liu et al., 2024c) with rank $= 8$ and dora_alpha $= 32$. The model is optimised using the Adam optimizer with a learning rate of $2 \times 10^{-5}$. A linear warmup is applied for the first $5\%$ of the training steps, followed by a cosine learning rate decay schedule. The model is trained for a total of 10 epochs. In each training step, both understanding and generation tasks were optimised concurrently, with a batch size of 6 for understanding tasks and 1 for generation tasks. Experiments were conducted on NVIDIA H20-140G GPUs, utilising the DeepSpeed ZeRO stage 2 optimisation and bf16 precision for training.

For understanding tasks, the input time series is directly flattened from $\mathbb{R}^{T' \times N}$ to $\mathbb{R}^{NT' \times 1}$. For generation tasks, the preprocessing strategy depends on the number of channels in the output signal. If the input signal is multi-channel but the output signal contains only a single channel, the input is flattened in the same manner as for understanding tasks. If the input and output signals have the same number of channels, prediction is performed on a per-channel basis without flattening.

# D  MORE DETAILS OF SCITS

This section provides more details and statistics for the SciTS benchmark. Table 5 lists the number of instances, range of length, frequency and number of channels for each discipline.

Table 5: Discipline-level statistics of SciTS.

| Discipline | # Instances | Min Len. | Max Len. | Min Freq. (Hz) | Max Freq. (Hz) | Dim. Range |
|---|---|---|---|---|---|---|
| Astronomy | 2583 | 24 | 6.4e4 | 1.2e-5 | 1.6e4 | 1 |
| Earth Science | 3080 | 6.0e3 | 6.0e3 | 1.0e2 | 1.0e2 | 3 |
| Bioacoustics | 4976 | 2.9e3 | 1.1e7 | 2.2e4 | 9.6e4 | 1 |
| Meteorology | 3976 | 16 | 3.4e2 | 2.8e-4 | 2.8e-4 | 1~5 |
| Economics | 1946 | 50 | 7.6e2 | 1.2e-5 | 1.1e-3 | 1~5 |
| Neuroscience | 7146 | 68 | 4.0e3 | 1.0e2 | 5.0e3 | 1~58 |
| Energy | 8501 | 48 | 2.6e2 | 2.8e-4 | 2.5e-1 | 1 |
| Physiology | 9824 | 9 | 5.0e3 | 1.3e2 | 5.0e2 | 1~12 |
| Urbanism | 3492 | 14 | 2.6e2 | 2.8e-4 | 2.8e-4 | 1 |
| Manufacturing | 4400 | 1.2e3 | 1.6e5 | 1.2e4 | 1.6e4 | 1 |
| Radar | 2243 | 33 | 3.2e2 | 3.2e6 | 1.0e7 | 2 |
| Math | 1856 | 1.0e2 | 1.0e2 | 1.0e2 | 1.0e2 | 3 |

## D.1  TASKS DESCRIPTION

Table 6 presents the tasks associated with each discipline, along with their IDs and types. We give a detailed description of each task below.

**Astronomy**. In the field of astronomy, researchers analyse celestial phenomena using diverse data sources. Three tasks are involved:

- **ASU01-Gravitational wave anomaly detection**: Identifying unusual patterns in detector data that may indicate gravitational wave signals among noise and instrumental artifacts. The input to this task consists of time series data and text, and the output is text.
- **ASG02-Gravitational wave event localisation**: Verifying candidate signals and determine the occurrence time of gravitational waves. The input to this task consists of time series data and text, and the output is text.
- **ASU03-Light curve classification**: Categorising celestial objects by their brightness variations over time to identify phenomena such as supernovae, variable stars, and exoplanetary transits. The input for this task includes time series data and text, with the output being text.

**Earth Science**. In the field of Earth science, researchers analyse seismic waveform data from global networks to monitor and understand subsurface activities. Two tasks are involved:

- **EAU01-Earthquake anomaly detection**: Identifying unusual signals in seismic data that do not conform to normal noise or typical earthquake waveform patterns, which may indicate abnormal geophysical processes or potential disaster precursors. The input to this task consists of three-channel time series data and text, and the output is text.
- **EAG02-Earthquake event localisation**: Automatically identifying and locate signals produced by real earthquakes, such as P waves and S waves, in continuous seismic records, and accurately determine their origin time, hypocentral location, and magnitude. The input for this task includes three-channel time series data and text, with the output being text.

**Bioacoustics**. In the field of bioacoustics, scientists monitor and study wildlife by collecting and analysing sound data from the environment. Three tasks are involved:

Table 6: Task overview of SciTS.

| Discipline | Task Brief | Task ID | Task Type |
|---|---|---|---|
| Astronomy | Gravitational wave anomaly detection | ASU01 | Anomaly detection |
| | Gravitational wave event localisation | ASG02 | Event localisation |
| | Light curve classification | ASU03 | Classification |
| Earth Science | Earthquake anomaly detection | EAU01 | Anomaly detection |
| | Earthquake event localisation | EAG02 | Event localisation |
| Bioacoustics | Birds vocalisation classification | BIU01 | Classification |
| | Animal vocalisation classification | BIU02 | Classification |
| | Marmoset vocalisation classification | BIU03 | Classification |
| Meteorology | Weather anomaly event detection | MEU01 | Anomaly detection |
| | Rainfall anomaly event detection | MEU02 | Anomaly detection |
| | Temperature forecasting | MEG03 | Forecasting |
| | Temperature MCQ | MEU04 | MCQ |
| Economics | Stock closing price forecasting | ECG01 | Forecasting |
| | Stock price forecasting | ECG02 | Forecasting |
| | Stock MCQ | ECU03 | MCQ |
| Neuroscience | Depressive disorder detection | NEU01 | Anomaly detection |
| | EEG pattern classification | NEU02 | Classification |
| | EEG signal forecasting | NEG03 | Forecasting |
| | EEG signal imputation | NEG04 | Imputation |
| | Motor imagery classification | NEU05 | Classification |
| | Sleep staging classification | NEU06 | Classification |
| Energy | Generate sensor signal trend based on description | ENG01 | Synthesis |
| | Transformer sensor signal forecasting | ENG02 | Forecasting |
| | Electricity load forecasting | ENG03 | Forecasting |
| | Comprehensive electricity forecasting | ENG04 | Forecasting |
| | Comprehensive electricity imputation | ENG05 | Imputation |
| Physiology | ECG status classification | PHU01 | Classification |
| | Physiological signal forecasting | PHG02 | Forecasting |
| | Physiological signal imputation | PHG03 | Imputation |
| | ECG anomaly detection | PHU04 | Anomaly detection |
| | Gait freezing anomaly detection | PHU05 | Anomaly detection |
| | Human activity classification | PHU06 | Classification |
| Urbanism | Traffic flow forecasting | URG01 | Forecasting |
| | Pedestrian flow forecasting | URG02 | Forecasting |
| | Pedestrian flow imputation | URG03 | Imputation |
| | Traffic flow anomaly detection | URU04 | Anomaly detection |
| | Traffic volume forecasting | URG05 | Forecasting |
| Manufacturing | Bearings fault location detection from vibration signals | MFU01 | Classification |
| | Bearings fault size detection from vibration signals | MFU02 | Classification |
| | Machine malfunction detection from operational recordings | MFU03 | Anomaly detection |
| Radar | Coding scheme classification | RAU01 | Classification |
| | Modes and modulation classification | RAU02 | Classification |
| Math | Evolution of chaotic system forecasting | MAG01 | Forecasting |

- **BIU01-Birds vocalisation classification**: Identifying bird species present in audio recordings, typically by analysing their distinctive calls or songs. The input to this task consists of time series data and text, and the output is text.

- **BIU02-Animal vocalisation classification**: Recognising sounds from a broader range of animal groups, such as mammals, insects, and amphibians. The input to this task consists of time series data and text, and the output is text.

- **BIU03-Marmoset vocalisation classification**: Specialized in classifying the vocalizations of marmosets, a type of small New World monkey, often used to study their complex social communication. The input to this task consists of time series data and text, and the output is text.

**Meteorology**. In the field of meteorology, researchers analyse multi-source meteorological data to understand and predict atmospheric phenomena. Four tasks are involved:

- **MEU01-Weather anomaly event detection**: Identifying extreme weather events that deviate from normal climate patterns, such as cold waves, heatwaves, or persistent heavy rainfall. The input to this task consists of time series data and text, and the output is text.

- **MEU02-Rainfall anomaly event detection**: Detecting rainfall patterns that significantly depart from historical norms, supporting flood early warning and water resource management. The input to this task consists of five-channel time series data and text, and the output is text.

- **MEG03-Temperature forecasting**: Forecasting temperature changes over specific future time periods, forming the basis for daily and agricultural meteorological services. The input to this task consists of single-channel time series data and text, and the output is single-channel time series data with a maximum length of 72.

- **MEU04-Temperature MCQ**: Systematically analysing temperature data, identify trends or potential anomalies in the data, and provide a basis for subsequent decision-making. The input to this task consists of time series data and text, and the output is text.

**Economics**. In the field of economics and finance, researchers and practitioners analyse market data to predict price movements and ensure data quality. Three tasks are involved:

- **ECG01-Stock closing price forecasting**: Predicting the final price of a stock at the end of a specific trading day, a key reference for investment decisions. The input to this task consists of five-channel time series data and text, and the output is single-channel time series data.

- **ECG02-Stock price forecasting**: Predicting the stock price or its movement over a future time horizon. The input to this task consists of single-channel time series data and text, and the output is single-channel time series data.

- **ECU03-Stock MCQ**: Systematically analysing stock market data to identify and rectify erroneous or anomalous records, ensuring data accuracy and reliability. The input for this task includes time series data and text, with the output being text.

**Neuroscience**. In the field of neuroscience, researchers analyse electroencephalogram (EEG) signals to understand brain function, diagnose neurological disorders, and develop brain-computer interfaces. Six tasks are involved:

- **NEU01-Depressive disorder detection**: Detecting changes in brain activity related to depressive episodes by analysing the characteristic patterns of EEG signals. The input to this task consists of multichannel time series data and text, and the output is text.

- **NEU02-EEG pattern classification**: Recognising different types of waveforms in EEG signals, commonly used in the diagnosis of epilepsy and other disorders. The input to this task consists of multichannel time series data and text, and the output is text.

- **NEG03-EEG signal forecasting**: Forecasting the EEG signals of various patients, including amyotrophic lateral sclerosis (ALS), major depressive disorder (MDD), ADHD, and OCD, can help understand disease progression and evaluate treatment outcomes. The input to this task consists of single-channel time series data and text, and the output is single-channel time series data with a maximum length of 32.

- **NEG04-EEG signal imputation**: Filling in missing data caused by artifacts or equipment issues in EEG recordings of various patients, including amyotrophic lateral sclerosis (ALS), major depressive disorder (MDD), ADHD, and OCD, to ensure data integrity. The input to this task consists of single-channel time series data and text, and the output is single-channel time series data with a maximum length of 12.

- **NEU05-Motor imagery classification**: Recognising unique brain patterns generated by different movement intentions. The input to this task consists of multi-channel time series data and text, and the output is text.

- **NEU06-Sleep staging classification**: Automatically classifying sleep into different stages based on EEG signal characteristics, used for sleep quality assessment and sleep disorder diagnosis. The input to this task consists of single-channel time series data and text, and the output is text.

**Energy**. In the field of energy, researchers analyse various types of energy data to optimize grid dispatch, improve energy efficiency, and ensure system stability. Five tasks involved:

- **ENG01-Generate sensor signal trend based on description**: Using ETT (Electricity Transformer Temperature) description to forecast the temperature of power transformers in the future in reverse, for analysing temperature trends and diagnosing anomalies. The input to this task is text, and the output is single-channel time series data with a maximum length of 96.

- **ENG02-Transformer sensor signal forecasting**: Forecasting the future temperature of power transformers, which is crucial for preventing equipment overheating, scheduling maintenance, and ensuring the safe operation of the power grid. The input to this task consists of single-channel time series data and text, and the output is single-channel time series data with a length of 96 for short-term forecasting and 720 for long-term forecasting.

- **ENG03-Electricity load forecasting**: Forecasting future electricity demand (load), which is fundamental to power system planning, dispatching, and energy trading. The input to this task consists of single-channel time series data and text, and the output is single-channel time series data with a maximum length of 48.

- **ENG04-Comprehensive electricity forecasting**: Forecasting future changes in electricity based on the demand, generation, or consumption of electricity in a certain city, and support the fine planning of the city's energy. The input to this task consists of single-channel time series data and text, and the output is single-channel time series data with a maximum length of 32.

- **ENG05-Comprehensive electricity imputation**: Filling in missing values in electricity demand, production, or consumption data caused by communication failures or equipment issues to ensure data integrity and achieve accurate analysis and prediction. The input to this task consists of single-channel time series data and text, and the output is single-channel time series data with a maximum length of 12.

**Physiology**. In the field of physiology, researchers analyse human physiological signals and health data to study bodily functions, monitor health status, and assist in disease management. Six tasks are involved:

- **PHU01-ECG status classification**: Classifying heart rhythm types or abnormal cardiac states by analysing electrical activity waveforms to assist in the diagnosis of cardiovascular diseases. The input to this task consists of 12-channel time series data and text, and the output is text.

- **PHG02-Physiological signal forecasting**: Forecasting future changes in physiological status based on historical health indicators, providing a basis for personalized health management and disease prevention. The input to this task consists of single-channel time series data and text, and the output is single-channel time series data with a maximum length of 32.

- **PHG03-Physiological signal imputation**: Filling in missing values in health monitoring data caused by device failures or monitoring interruptions to ensure data integrity for accurate analysis. The input to this task consists of single-channel time series data and text, and the output is single-channel time series data with a maximum length of 12.

- **PHU04-ECG anomaly detection**: Identifying abnormal segments that deviate from normal patterns in ECG signals, enabling timely warning of potential heart issues. The input to this task consists of single-channel time series data and text, and the output is text.

- **PHU05-Gait freezing anomaly detection**: Detecting sudden "freezing" episodes (temporary movement cessation) during walking in patients with Parkinson's disease, supporting rehabilitation assessment and intervention. The input to this task consists of single-channel time series data and text, and the output is text.

- **PHU06-Human activity classification**: Recognising types of daily human activities (e.g., walking, sitting, climbing stairs) using data from accelerometers and gyroscopes, for movement monitoring and health evaluation. The input to this task consists of single-channel time series data and text, and the output is text.

**Urbanism**. In the field of urbanism, researchers analyse traffic and pedestrian data to optimize urban planning, improve travel efficiency, and ensure public safety. Five tasks involved:

- **URG01-Traffic flow forecasting**: Forecasting future changes in vehicle flow on road networks during specific time periods, serving as the core basis for traffic management and signal control. The input to this task consists of single-channel time series data and text, and the output is single-channel time series data with a maximum length of 24.

- **URG02-Pedestrian flow forecasting**: Forecasting future pedestrian flow and distribution in specific areas (such as around business districts, subway stations, and large venues) to support public safety management and business decisions. The input to this task consists of single-channel time series data and text, and the output is single-channel time series data with a maximum length of 32.

- **URG03-Pedestrian flow imputation**: Filling in missing values in pedestrian monitoring data caused by equipment failures, network interruptions, or other issues to ensure the accuracy and continuity of data analysis. The input to this task consists of single-channel time series data and text, and the output is single-channel time series data.

- **URU04-Traffic flow anomaly detection**: Identifying abnormal conditions that deviate from normal patterns in real-time traffic data, such as traffic jams, accidents, or road closures, to enable timely warning and guidance. The input to this task consists of single-channel time series data and text, and the output is text.

- **URG05-Traffic volume forecasting**: Forecasting the total number of vehicles that will pass through a specific road segment or intersection in the future, which is fundamental to transportation planning, infrastructure construction, and traffic demand management. The input to this task consists of single-channel time series data and text, and the output is single-channel time series data with a maximum length of 7.

**Manufacturing**. In the field of manufacturing, researchers and engineers analyse industrial sensor data to enable equipment health management and intelligent maintenance. Two tasks are involved:

- **MFU01-Bearings fault location detection from vibration signals**: Classifying bearings health status or fault types by analysing signals such as bearing vibrations, which is a key technology for predictive maintenance. The input to this task consists of single-channel time series data and text, and the output is text.

- **MFU02-Bearings fault size detection from vibration signals** : Detecting the size of fault through the analysis of vibration signals from industrial bearings (including machine tools, pumps, and fans) helps identify abnormal states deviating from normal modes. The input to this task consists of single-channel time series data and text, and the output is text.

- **MFU03-Machine malfunction detection from operational recording**: Identifying abnormal patterns (which may indicate potential equipment failures) automatically relies on analysing various data generated while industrial equipment is in operation. The input to this task consists of single-channel time series data and text, and the output is text.

**Radar**. In the field of radar, researchers analyse radar echo signals to achieve target detection, recognition, and environmental perception. Two tasks are involved:

- **RAU01-Coding scheme classification**: Classifying radar data segments to identify ground targets or terrain types. The input to this task consists of two-channel time series data and text, and the output is text.

- **RAU02-Modes and modulation classification**: Classifying raw or processed signals received by radar to identify their modulation type, source, or target attributes. The input to this task consists of two-channel time series data and text, and the output is text.

Table 7: Task-level statistics of SciTS.

| Discipline | Task ID | # Instances | Input TS Length Range | Generation TS Length Range | Frequency Range(Hz) | Channel Range | Data Source |
|---|---|---|---|---|---|---|---|
| Astronomy | ASU01 | 1024 | 6.4e4 | / | 1.6e4 | 1 | Simulation |
| | ASG02 | 535 | 6.4e4 | 1 | 1.6e4 | 1 | Simulation |
| | ASU03 | 1024 | 24~1.3e3 | / | 1.2e-5 | 1 | LEAVES (Fei et al., 2024) |
| Earth Science | EAU01 | 2048 | 6.0e3 | / | 1.0e2 | 3 | STEAD (Mousavi et al., 2019) |
| | EAG02 | 1032 | 6.0e3 | 2 | 1.0e2 | 3 | STEAD (Mousavi et al., 2019) |
| Bioacoustics | BIU01 | 1602 | 2.9e3~4.2e6 | / | 3.2e4 | 1 | Powdermill (Chronister et al., 2021) |
| | BIU02 | 2474 | 1.7e4~1.1e7 | / | 2.2e4 | 1 | iNaturalist (Chasmai et al., 2024) |
| | BIU03 | 900 | 6.2e4~2.4e5 | / | 9.6e4 | 1 | MarmAudio (Lamothe et al., 2025) |
| Meteorology | MEU01 | 1256 | 16 | / | Unknown | 1 | TS MQA (Kong et al., 2025) |
| | MEU02 | 1131 | 24 | / | 2.8e-4 | 5 | TIMECAP (Lee et al., 2025) |
| | MEG03 | 1176 | 1.7e2~3.4e2 | 24~72 | 2.8e-4 | 1 | MTbench (Chen et al., 2025b) |
| | MEU04 | 413 | 1.7e2~3.4e2 | / | 2.8e-4 | 1 | MTbench (Chen et al., 2025b) |
| Economics | ECG01 | 1260 | 50 | 3 | 1.2e-5 | 5 | FinMultiTime (Xu et al., 2025b) |
| | ECG02 | 383 | 1.2e2~7.6e2 | 11~2.0e2 | 1.1e-3 | 1 | MTbench (Chen et al., 2025b) |
| | ECU03 | 303 | 1.1e2~4.0e2 | / | 1.1e-3 | 1 | MTbench (Chen et al., 2025b) |
| Neuroscience | NEU01 | 1000 | 2.5e3 | / | 2.6e2 | 18~20 | MDD (Mumtaz, 2016) |
| | NEU02 | 789 | 2.5e2~4.0e3 | / | 2.5e2 | 23 | TUEV (Obeid & Picone, 2016) |
| | NEG03 | 360 | 68~2.5e2 | 8~32 | 2.6e2 ~ 5.0e2 | 1 | TS MQA (Kong et al., 2025) |
| | NEG04 | 373 | 97~2.6e2 | 4~12 | 2.6e2 ~ 5.0e2 | 1 | TS MQA (Kong et al., 2025) |
| | NEU05 | 3600 | 1.0e3 | / | 2.5e2 | 58 | WBCIC SHU (Yang et al., 2025b) |
| | NEU06 | 1024 | 3.0e3 | / | 1.0e2 | 1 | Sleep (Kemp et al., 2000) |
| Energy | ENG01 | 2700 | / | 24~96 | 2.8e-4 | 1 | textETT (Ge et al., 2025) |
| | ENG02 | 5570 | 96 | 96~7.2e2 | 2.8e-4 | 1 | ETT (Wu et al., 2021) |
| | ENG03 | 100 | 48 | 48 | 5.6e-4 | 1 | NewsForecast (Wang et al., 2024b) |
| | ENG04 | 66 | 1.0e2~2.5e2 | 8~32 | 2.8e-4 ~ 2.5e-1 | 1 | TS MQA (Kong et al., 2025) |
| | ENG05 | 65 | 2.6e2 | 4~12 | 2.8e-4 ~ 2.5e-1 | 1 | TS MQA (Kong et al., 2025) |
| Physiology | PHU01 | 2090 | 5.0e3 | / | 5.0e2 | 12 | PTB XL (Wagner et al., 2020) |
| | PHG02 | 1517 | 67~2.5e2 | 8~32 | 1.3e2 ~ 2.6e2 | 1 | TS MQA (Kong et al., 2025) |
| | PHG03 | 1511 | 97~2.6e2 | 4~12 | 1.3e2 ~ 2.6e2 | 1 | TS MQA (Kong et al., 2025) |
| | PHU04 | 1006 | 64 | / | 5.0e2 | 1 | TS MQA (Kong et al., 2025) |
| | PHU05 | 1882 | 9 | / | 64 | 1 | TS MQA (Kong et al., 2025) |
| | PHU06 | 1818 | 30 | / | 20 | 1 | TS MQA (Kong et al., 2025) |
| Urbanism | URG01 | 43 | 24 | 24 | 2.8e-4 | 1 | NewsForecast (Wang et al., 2024b) |
| | URG02 | 62 | 1.1e2~2.5e2 | 8~32 | 2.8e-4 | 1 | TS MQA (Kong et al., 2025) |
| | URG03 | 65 | 1.3e2~2.6e2 | 4~12 | 2.8e-4 | 1 | TS MQA (Kong et al., 2025) |
| | URU04 | 122 | 16 | / | Unknown | 1 | TS MQA (Kong et al., 2025) |
| | URG05 | 3200 | 14~21 | 3~7 | 2.8e-4 | 1 | MetroTraffic (Wang et al., 2025a) |
| Manufacturing | MFU01 | 400 | 1.2e3~4.8e3 | / | 1.2e4 | 1 | CWRU (Center) |
| | MFU02 | 400 | 1.2e3~4.8e3 | / | 1.2e4 | 1 | CWRU (Center) |
| | MFU03 | 3600 | 1.6e5 | / | 1.6e4 | 1 | MIMII Due (Tanabe et al., 2021) |
| Radar | RAU01 | 963 | 33~3.2e2 | / | 3.2e6 | 2 | RadSeg (Huang et al., 2024) |
| | RAU02 | 1280 | 1.3e2 | / | 1.0e7 | 2 | RadarComm (Jagannath & Jagannath, 2021) |
| Math | MAG01 | 1856 | 1.0e2 | 20 | 1.0e2 | 3 | Chaotic (Gilpin, 2023) |

**Mathematics**. In the field of mathematics, researchers analyse the behavior of chaotic systems to understand their dynamic properties and make predictions. One task involved:

- **MAG01-Evolution of chaotic system forecasting**: Performing short-term predictions for chaotic systems that are driven by underlying ordinary differential equations (ODEs) governing their dynamics (such as the Lorenz system). These systems exhibit extreme sensitivity to initial conditions and complex nonlinear behaviors. The input to this task consists of single-channel time series data and text, and the output is text with a length of 20.

Table 7 presents task-level statistics, including the ranges of input and target time series lengths (where applicable) and the corresponding data sources.

## D.2 DATASET DESCRIPTION

This section provides description of the datasets contained in SciTS.

**GWOSC GW Event.** Since the number of observed gravitational events is quite limited. We constructed GWOSC GW Event dataset by simulation, following prior work in astronomy (Dax et al., 2021; 2023; 2025). To construct synthetic data that closely match real observations, we generated gravitational-wave templates via numerical relativity simulations and injected them into strain data

downloaded from the GWOSC platform[5]. Unlike the traditional approach of generating noise from a fixed power spectral density (PSD), this method leverages real detector noise segments to enrich noise diversity. Specifically, we applied data-quality and injection masks to exclude low-quality or contaminated intervals, and extracted 4-second noise segments (sampled at 16 kHz) starting 16 s after each event to form a noise pool. A template bank was built through a grid search over typical binary black hole mass ranges. For each template, two noise segments were randomly drawn and scaled to different matched-filter SNRs by adjusting the amplitude ratio. To mitigate truncation artifacts, windowing was applied during waveform-noise superposition, and the final data were whitened to produce realistic and training-ready synthetic datasets.

**LEAVES** (Fei et al., 2024) is a large-scale light-curve dataset designed for the automatic classification of variable stars. It is constructed by merging open data from three major astronomical surveys: the All-Sky Automated Survey for SuperNovae (ASAS-SN), Gaia, and the Zwicky Transient Facility (ZTF). The dataset comprises 977,953 light curves from variable stars and 134,592 from nonvariable sources. All variable stars are annotated with seven superclasses: Eclipsing binaries, Rotational variables, RR Lyrae, Cepheids, Long-period variables, Delta Scuti, and Nonvariable.

**STEAD** (Mousavi et al., 2019) is a large-scale global dataset containing 1.2M labelled waveforms, 19k hours of data, and 450k earthquakes that occurred between January 1984 and August 2018. All waveforms from the dataset have three channels, each of 1-minute length with a sampling rate of 100Hz. We process the three-channel data by calculating its Euclidean norm, which transforms the data into a single-channel waveform representing the total amplitude of the ground motion.

**Powdermill** (Chronister et al., 2021) is the first strongly labelled bird soundscape recording dataset (with information on timing, frequency, and species). Collected at the Powdermill Nature Reserve in Pennsylvania, northeastern United States, the dataset includes dawn chorus recordings captured by acoustic recorders between April and July 2018. Continuous recordings were made over four days during the study period, covering 48 species and 16,052 annotations.

**iNaturalist Sounds dataset (iNatSounds)** (Chasmai et al., 2024) is a large-scale collection of 230,000 audio recordings capturing sounds from over 5,500 species, including birds, mammals, insects, reptiles, and amphibians. Each recording in the dataset varies in length and includes a single species annotation.

**MarmAudio** (Lamothe et al., 2025) is a large-scale dataset of common marmoset vocalizations, recorded over 40 months at a high sampling rate of 96 kHz in a soundproofed animal facility room containing three cages ($\approx$20 marmosets). The dataset comprises more than 800,000 short audio segments ($\approx$253 hours), capturing the full vocal repertoire observed during the recording period. A subset of approximately 215,000 calls ($\approx$72 hours) is annotated with six vocalization types—Infant cry, Phee, Seep, Trill, Tsik, and Twitter—providing rich supervision for downstream analysis.

**Time MQA** (Kong et al., 2025) is a large-scale dataset designed for time series multi-task question answering, constructed from a wide range of publicly available time series benchmarks. It spans over twelve domains, including healthcare, finance, and energy. The dataset comprises approximately 200,000 question-answer pairs, covering five tasks: forecasting, imputation, anomaly detection, classification, and open-ended reasoning. Each entry is enhanced with contextual text, such as background information and feature descriptions, to provide rich supervision for both numerical analysis and advanced reasoning tasks.

**TimeCAP** (Lee et al., 2025) dataset in the weather domain is a sub-dataset that consists of hourly time series data on temperature, humidity, air pressure, wind speed, and wind direction in New York (NY), San Francisco (SF), and Houston (HS). Given the last 24 hours of time series data, the task is to predict whether it will rain in the next 24 hours.

**Multimodal Time Series Benchmark (MTBench)** (Chen et al., 2025b) is a cross-domain dataset, covering two domains: weather and finance. It comprises paired time-series and textual data, including financial news with corresponding stock price movements and weather reports aligned with historical temperature records. The richness of MTBench enables the formulation of diverse tasks that require a deep understanding of both text and time-series data, including time-series forecasting, semantic and technical trend analysis, and news-driven QA.

---

[5]https://gwosc.org/eventapi/html/

**FinMultiTime** (Xu et al., 2025b) is a dataset in the field of Economy, which aligns four distinct modalities: financial news, structured financial tables, K-line technical charts, and stock price time series across both the S&P 500 and HS 300 universes. Covering 5,105 stocks from 2009 to 2025 in the United States and China. This dataset provides minute-level, daily, and quarterly resolutions, thus capturing short, medium, and long-term market signals with high fidelity.

**MDD** (Mumtaz, 2016) is a dataset that includes 33 patients diagnosed with major depressive disorder (MDD) according to DSM-IV criteria without psychotic symptoms (18 females, mean age = 40.33, SD = 12.861) and 30 age-matched healthy controls (9 females, mean age = 38.23, SD = 15.64). Clinical assessments comprised the Beck Depression Inventory-II (BDI-II) and the Hospital Anxiety and Depression Scale (HADS), while neurophysiological data consisted of resting-state EEG recordings. EEG was collected with a 19-channel cap based on the 10-20 system, using linked-ear (LE) reference initially and later re-referenced to infinity reference (IR).

**TUEV (The TUH EEG Events Corpus)** (Obeid & Picone, 2016) is a subset of TUEG, which focuses on clinical electroencephalogram (EEG) recordings conducted at Temple University Hospital (TUH) from 2002 to 2013 (and beyond). TUEV contains annotations of EEG segments as one of six classes: (1) spike and sharp wave (SPSW), (2) generalized periodic epileptiform discharges (GPED), (3) periodic lateralized epileptiform discharges (PLED), (4) eye movement (EYEM), (5) artifact (ARTF) and (6) background (BCKG).

**WBCIC SHU** (Yang et al., 2025b) dataset is a comprehensive motor imagery brain computer interface dataset comprising data from 62 subjects across three recording sessions. This dataset includes two paradigms: the first involves upper limb movements with left and right hand-grasping, with 51 subjects participating; the second adds a foot-hooking task, involving 11 subjects.

**SleepEDF** (Kemp et al., 2000) is a comprehensive dataset containing 153 overnight polysomnography (PSG) recordings from 78 subjects. Each recording includes signals of two-channel EEG (Fpz-Cz and Pz-Cz) and one EOG, sampled at 100Hz. We process the data by utilizing only the Fpz-Cz EEG channel, excluding the 'MOVEMENT' and 'UNKNOWN' stages, and combining the N3 and N4 stages into one sleep stage N3, to conform to the latest AASM standard, yielding the five target sleep stages: W, N1, N2, N3, and REM. The EEG signals are processed with a 0.5-45Hz bandpass Butterworth filter to remove background noise.

The **textETT** (Ge et al., 2025) dataset is an augmented version of the ETT dataset, where each time series fragment is paired with a textual description. In this dataset, the input is a natural language caption, and the output is the corresponding time series segment. Following the settings introduced in the original paper, we use ETTh1 and construct fragment-level datasets with lengths of 24, 48, and 96.

**ETT** (Wu et al., 2021) is a dataset that contains measurements from electricity transformers, including six load features and oil temperature. We use the ETTh1 subset, which is hourly recorded from July 2016 to June 2018. Following the standard protocol, we split the dataset chronologically into training, validation, and test sets with a ratio of 6:2:2. In Sci-TS, we focus on the OT (oil temperature) column for univariate forecasting with a context length of 96 and prediction lengths of 96 and 720, allowing us to assess model performance on both short-term and long-term predictions.

**NewsForecast** (Wang et al., 2024b) is a multi-domain dataset that combines time series data from multiple human-driven domains with aligned news and contextual information. The time series covers four domains: Traffic (hourly traffic volume), Exchange (daily exchange rates), Bitcoin (daily Bitcoin price), and Electricity (half-hourly electricity demand from the Australian Energy Market Operator, AEMO). To enrich the forecasting task with external signals, the dataset incorporates related news. Part of the news content is drawn from the GDELT database, which tracks global events in over 100 languages, while additional domain-specific news is collected from sources such as News Corp Australia and Yahoo Finance.

**PTB XL** (Wagner et al., 2020) dataset consists of 21,837 clinical 12-lead ECG recordings (10-second each) from 18,885 patients, sampled originally at 500 Hz (also provided downsampled to 100 Hz). The records are multi-label annotated with up to 71 SCP-ECG diagnostic/form/rhythm statements, with hierarchy into super- and subclass diagnostic labels, and include metadata like age, sex, signal quality, recording device and site.

**MetroTraffic (Metro Interstate Traffic Volume)** (Wang et al., 2025a) is a dataset that contains hourly westbound traffic volumes on Interstate 94 between Minneapolis and St. Paul, MN, from 2012 to 2018, including 63 holidays. In Sci TS, we set two forecast modes: forecasting the next 3 days from the past 14 days and forecasting the next 7 days from the past 21 days. By sliding on the raw sequence data according to different prediction lengths, we finally obtained approximately 25k pieces of data.

**CWRU (Case Western Reserve University)** (Center) bearing dataset contains vibration data from normal and artificially faulted ball bearings, collected using a 2 HP Reliance Electric motor. Faults (created by electro-discharge machining) include defects on the inner raceway, the rolling element (ball), and the outer raceway, with fault diameters of 0.007", 0.014", 0.021", and 0.028". Vibration is measured at multiple sensor locations: drive end (DE), fan end (FE), and base (BA), under motor loads from 0 to 3 HP and motor speeds around 1720-1797 RPM.

**MIMII DUE** (Tanabe et al., 2021) is a dataset that consists of the normal and abnormal operating sounds of five different types of industrial machines. The data for each machine type includes six subsets(three utilized in Sci-TS) called "sections", and each section roughly corresponds to a single product. Each section consists of data from two domains, i.e., the source domain and the target domain, with different conditions such as operating speed and environmental noise.

**RadSeg (Radar Segmentation Dataset)** (Huang et al., 2024) is a synthetic radar dataset designed for building semantic segmentation models for radar activity recognition. Unlike existing radio classification datasets that only provide signal-wise annotations for short and isolated IQ sequences, RadSeg provides sample-wise annotations for interleaved radar pulse activities that extend across a long time horizon. RadSeg contains pulsed radar signals at varying signal-to-noise ratios (SNRs) between -20 and 20 dB with a resolution of 0.5 dB.

**RadarComm** (Jagannath & Jagannath, 2021) is a wireless signal dataset. The lack of existing multitask labelled datasets for machine learning for wireless communication is the prime motivation urging this release. RadarCommDataset is the first of its kind, a multitask labelled dataset released to help the research community advance machine learning for wireless communication. The dataset contains radar and communication waveforms.

**Chaotic** (Gilpin, 2023) datasets is a collection of multivariate time series generated from 135 well-known chaotic systems, each numerically integrated to produce sequences of length 10,000 with highly coupled dynamics. To account for integration precision, the dataset is divided into three categories: coarse, medium, and fine. We select the coarse subset and, after filtering out anomalous cases, retain 116 systems. Each task uses a context length of 100 to forecast the next 20 time steps.

## E DESCRIPTION AND CONFIGURATION OF THE ASSESSED MODELS

This section provides descriptions and configurations of assessed models, including text-only LLMs, multimodal LLMs, and unified time-series models. Our preliminary experiments suggest that reasoning provides little benefit for time-series tasks in SciTS for the assessed models. Given resource constraint, we set the `reasoning_effort` parameter to `minimal` for all evaluations.

### E.1 TEXT LLMS

1. **GPT-4.1-mini**[6]. A lightweight version of OpenAI's GPT-4 series, optimized for efficiency while maintaining strong reasoning and language understanding capabilities, with a context length of 1M tokens.

2. **Gemini-2.5-Flash**[7]. The most advanced model from Google DeepMind's Gemini family, designed for fast inference with competitive performance across diverse language tasks, with a context length of 1M tokens.

3. **DeepSeek-V3** (Liu et al., 2024a)[8]. An advanced open-source LLM developed by DeepSeek, with a context length of 128K tokens.

---

[6] https://platform.openai.com/docs/models/gpt-4.1-mini
[7] https://ai.google.dev/gemini-api/docs/models?gemini-2.5-flash
[8] https://huggingface.co/deepseek-ai/DeepSeek-V3

4. **Llama3-8B** (Grattafiori et al., 2024)[9]. An open-source LLM from Meta's Llama 3 series, widely adopted for research applications, with a context length of 8k tokens.

5. **Qwen3-4B** (Yang et al., 2025a)[10]. A smaller version of Qwen3, providing lightweight inference while maintaining competitive performance, with a suggested context length of 128k tokens.

6. **Qwen3-8B** (Yang et al., 2025a)[11]. An open source LLM from Alibaba's Qwen series, with a suggested context length of 128k tokens.

### E.2    MULTIMODAL LLMS

1. **GPT-5-mini**[12]. A multimodal variant of OpenAI's most advanced GPT-5 family, with a context length of 400K tokens.

2. **Gemini-2.5-Flash**[13]. We also use it for evaluation with multimodal inputs since it is a native multimodal LLM.

3. **InternVL3.5-4B** (Wang et al., 2025c)[14]. A smaller 4B-parameter version of InternVL3.5.

4. **InternVL3.5-8B** (Wang et al., 2025c)[15]. An advanced open-source multimodal LLM from Shanghai AI Lab, with a context length of 32K tokens.

5. **Qwen2.5-VL-7B** (Bai et al., 2025b)[16]. A multimodal model from Alibaba's Qwen2.5 series, with a context length of 128K tokens.

### E.3    TIME SERIES MODELS

For time series models, we only incorporate models that support zero-shot inference. Therefore, foundation feature extraction models like PatchTST (Nie et al., 2023) are excluded. Among the baseline models listed below, **Moirai**, **Time-MOE**, and **Chronos** are limited to forecasting tasks. In contrast, **ChatTS** focuses solely on understanding tasks, while **UniTS** is a unified model capable of handling both understanding and generation tasks, including forecasting and imputation.

1. **Moirai** (Woo et al., 2024). A universal time series forecasting model builds on a masked encoder Transformer. It incorporates multi-patch projections to handle various data and supports multi-channel forecasting by flattening into a single sequence and using its specialized attention mechanism. We use Moirai-large (311M parameters), Moirai-base (91M parameters), Moirai-small (14M parameters) for evaluation. It supports multi-variate to univariate and univariate forecasting. We use univariate forecasting for multi-variate to multi-variate case.

2. **Time-MoE** (Shi et al., 2025).The first time series foundation model that incorporates a Mixture-of-Experts (MoE) architecture, scaling up to 2.4 billion parameters and trained from scratch. It employs point-wise tokenisation and leverages multi-resolution scheduling to produce forecasts at multiple scales simultaneously. The model is currently only capable of doing univariate time series forecasting. We use Time-MoE-base (50M parameters), Time-MoE-large (200M parameters) for evaluation. It only supports univariate forecasting. We use univariate forecasting for multi-variate to multi-variate case.

3. **Chronos** (Ansari et al., 2024). A family of pretrained time series forecasting models built on language model architectures. It transforms time series into discrete token sequences through scaling and quantization, and is trained using cross-entropy loss. During inference, Chronos generates probabilistic forecasts by sampling multiple future trajectories

---

[9]https://huggingface.co/meta-llama/Meta-Llama-3-8B

[10]https://huggingface.co/Qwen/Qwen3-4B

[11]https://huggingface.co/Qwen/Qwen3-8B

[12]https://platform.openai.com/docs/models/gpt-5-mini

[13]https://ai.google.dev/gemini-api/docs/models?gemini-2.5-flash

[14]https://huggingface.co/OpenGVLab/InternVL3_5-4B

[15]https://huggingface.co/OpenGVLab/InternVL3_5-8B

[16]https://huggingface.co/Qwen/Qwen2.5-VL-7B-Instruct

from the historical context. We use Chronos-bolt-mini (21M parameters) and Chronos-bolt-base (205M parameters) for evaluation. It only supports univariate forecasting. We use univairate forecasting for multi-variate to multi-variate case.

4. **ChatTS** (Xie et al., 2025). A universal time-series dialogue and understanding model constructed upon Large Language Models (LLMs) and equipped with a time-series encoder. It is capable of processing multivariate sequences (with a maximum support for 50-channel data input) and variable-length sequences, while supporting tasks such as trend analysis, clustering, and reasoning-based question answering. But it exclusively generates text output and does not support time-series generation tasks.

5. **UniTS** (Gao et al., 2024). A unified time series model that can handle a wide range of tasks within a single architecture[17]. It supports forecasting, anomaly detection, classification, and imputation, providing a general solution across diverse applications. It supports multivariate signal but requires input dimension to be the same as output dimension.

## F    FULL RESULTS ON SCITS

This section provides detailed results for each tasks. We define the following abbreviations for common failure reasons: (i) too long input/output sequence (TLS) where input length exceeds model's max context length or output prediction length exceeds model's max predict length; (ii) too many channels (TMC) where model doesn't support the number of input/output channels; (iii) instruction not followed (INF) where model does not generate output of required length or format. Understanding tasks are evaluated by accuracy and F1. Generation Tasks are evaluated by MAE, MAPE, success rate (SR). Not supported tasks are marked as "-". All results presented in percentage.

Table 8: Detailed results of SciTS (Astronomy and Earth Science).

| Domain | Astronomy | | | | | | Earth Science | | | |
|---|---|---|---|---|---|---|---|---|---|---|
| Task ID | ASU01 | | ASG02 | | ASU03 | | EAU01 | | EAG02 | |
| Metric | Acc | F1 | MAPE | SR | Acc | F1 | Acc | F1 | MAPE | SR |
| Large Language Models (Text input) | | | | | | | | | | |
| GPT-4.1-mini | 51.3 | 67.2 | 97.5 | 96.6 | 32.3 | 15.6 | 51.4 | 67.0 | 64.9 | 99.9 |
| Gemini2.5-Flash | 50.6 | 64.1 | 98.3 | 84.3 | 46.3 | 16.3 | 51.8 | 67.6 | 62.9 | 99.8 |
| DeepSeek-V3 | 48.0 | 1.1 | 99.4 | 17.0 | 32.9 | 12.3 | 53.8 | 40.2 | 86.6 | 96.1 |
| Llama3-8B | TLS | TLS | TLS | TLS | 8.3 | 5.1 | TLS | TLS | TLS | TLS |
| Qwen3-8B | TLS | TLS | TLS | TLS | 14.7 | 6.2 | 45.7 | 47.2 | 28.1 | 0.1 |
| Qwen3-4B | TLS | TLS | TLS | TLS | 47.2 | 15.9 | 47.2 | 15.9 | 76.5 | 0.1 |
| Multimodal Large Language Models (Text+Image input) | | | | | | | | | | |
| GPT-5 | 50.4 | 65.7 | 50.3 | 90.8 | 50.5 | 18.9 | 51.6 | 67.6 | 25.4 | 100.0 |
| Gemini2.5-Flash | 53.5 | 61.6 | 77.6 | 71.4 | 45.1 | 15.2 | 61.7 | 72.5 | 24.6 | 99.9 |
| InternVL3.5-8B | 55.7 | 41.7 | 50.8 | 30.3 | 2.9 | 1.0 | 76.4 | 81.0 | 2.2e2 | 99.9 |
| InternVL3.5-4B | 53.9 | 68.7 | 80.5 | 100.0 | 39.7 | 10.7 | 63.8 | 73.6 | 2.8e2 | 99.8 |
| Qwen2.5-VL-7B | 48.4 | 2.9 | 6.7 | 1.5 | 27.5 | 11.3 | 75.8 | 80.6 | 53.5 | 99.5 |
| Time Series Models | | | | | | | | | | |
| Moirai-large-311M | - | - | - | - | - | - | - | - | - | - |
| Moirai-base-91M | - | - | - | - | - | - | - | - | - | - |
| Moirai-small-14M | - | - | - | - | - | - | - | - | - | - |
| TimeMoE-base-50M | - | - | - | - | - | - | - | - | - | - |
| TimeMoE-large-200M | - | - | - | - | - | - | - | - | - | - |
| Chronos-bolt-mini-21M | - | - | - | - | - | - | - | - | - | - |
| Chronos-bolt-base-205M | - | - | - | - | - | - | - | - | - | - |
| ChaTS | TLS | TLS | TLS | TLS | 38.3 | 11.3 | 61.7 | 64.8 | 3.3 | 60.1 |
| UniTS | 52.2 | 68.6 | 3.3e6 | 100.0 | 11.5 | 7.8 | 49.6 | 0 | INF | INF |
| TimeOmni | 69.0 | 73.5 | 2.8 | 100.0 | 87.8 | 72.8 | 80.9 | 82.5 | 2.2 | 100.0 |

---

[17]https://github.com/mims-harvard/UniTS/releases/download/ckpt/units_x32_pretrain_checkpoint.pth

Table 9: Detailed results of SciTS (Bioacoustics and Meteorology).

| Domain | Bioacoustics | | | | | | Meteorology | | | |
|---|---|---|---|---|---|---|---|---|---|---|
| Task ID | BIU01 | | BIU02 | | BIU03 | | MEU01 | | MEU02 | |
| Metric | Acc | F1 | Acc | F1 | Acc | F1 | Acc | F1 | Acc | F1 |
| Large Language Models (Text input) | | | | | | | | | | |
| GPT-4.1-mini | 1.0 | 0.2 | 13.3 | 7.3 | 17.9 | 12.7 | 53.7 | 44.0 | 47.0 | 36.0 |
| Gemini2.5-Flash | 4.7 | 1.5 | 44.3 | 16.9 | 15.6 | 12.4 | 50.6 | 60.9 | 35.2 | 41.2 |
| DeepSeek-V3 | 0.1 | 0.0 | 2.3 | 0.9 | 16.9 | 5.8 | 49.0 | 59.3 | 52.9 | 34.8 |
| Llama3-8B | TLS | TLS | TLS | TLS | TLS | TLS | 48.7 | 65.3 | 27.8 | 43.2 |
| Qwen3-8B | TLS | TLS | TLS | TLS | TLS | TLS | 53.5 | 64.9 | 40.1 | 41.4 |
| Qwen3-4B | TLS | TLS | TLS | TLS | TLS | TLS | 51.3 | 67.1 | 73.4 | 0.0 |
| Multimodal Large Language Models (Text+Image input) | | | | | | | | | | |
| GPT-5 | 2.8 | 0.8 | 33.4 | 13.4 | 26.4 | 17.9 | 54.4 | 30.4 | 30.6 | 42.9 |
| Gemini2.5-Flash | 2.2 | 0.9 | 28.4 | 14.0 | 16.8 | 8.3 | 49.5 | 64.1 | 33.0 | 41.7 |
| InternVL3.5-8B | 3.3 | 0.1 | 7.1 | 3.2 | 16.8 | 10.8 | 48.9 | 65.4 | 51.7 | 33.4 |
| InternVL3.5-4B | 2.0 | 0.1 | 6.1 | 2.4 | 17.8 | 8.8 | 54.3 | 63.1 | 73.3 | 0.0 |
| Qwen2.5-VL-7B | 3.3 | 0.3 | 81.0 | 22.9 | 16.7 | 4.8 | 54.1 | 36.4 | 0.0 | 0.0 |
| Time Series Models | | | | | | | | | | |
| Moirai-large-311M | - | - | - | - | - | - | - | - | - | - |
| Moirai-base-91M | - | - | - | - | - | - | - | - | - | - |
| Moirai-small-14M | - | - | - | - | - | - | - | - | - | - |
| TimeMoE-base-50M | - | - | - | - | - | - | - | - | - | - |
| TimeMoE-large-200M | - | - | - | - | - | - | - | - | - | - |
| Chronos-bolt-mini-21M | - | - | - | - | - | - | - | - | - | - |
| Chronos-bolt-base-205M | - | - | - | - | - | - | - | - | - | - |
| ChaTS | TLS | TLS | TLS | TLS | TLS | TLS | 56.3 | 52.8 | 36.5 | 41.1 |
| UniTS | 6.1 | 0.6 | 82.1 | 18.9 | 16.7 | 4.8 | 48.7 | 0 | 73.4 | 0 |
| TimeOmni | 42.6 | 34.1 | 83.4 | 51.2 | 89.6 | 89.2 | 52.0 | 65.2 | 48.4 | 38.8 |

Table 10: Detailed results of SciTS (Meteorology and Economics).

| Domain | Meteorology | | | | Economics | | | | | |
|---|---|---|---|---|---|---|---|---|---|---|
| Task ID | MEG03 | | | MEU04 | ECG01 | | | ECG02 | | |
| Metric | MAE | MAPE | SR | Acc | MAE | MAPE | SR | MAE | MAPE | SR |
| Large Language Models (Text input) | | | | | | | | | | |
| GPT-4.1-mini | 3.9 | 42.1 | 49.6 | 55.9 | 22.3 | 193.1 | 89.8 | 3.5 | 3.7 | 89.8 |
| Gemini2.5-Flash | 4.3 | 62.2 | 57.9 | 53.3 | 0.7 | 2.3 | 65.2 | 3.0 | 3.1 | 65.2 |
| DeepSeek-V3 | 4.7 | 46.4 | 30.9 | 55.5 | 1.1 | 3.2 | 90.1 | 4.9 | 5.3 | 90.1 |
| Llama3-8B | INF | INF | INF | 39.0 | INF | INF | INF | INF | INF | INF |
| Qwen3-8B | 0.8 | 27.5 | 0.2 | 56.7 | INF | INF | INF | 1.0 | 6.4 | 1.3 |
| Qwen3-4B | INF | INF | INF | 55.7 | INF | INF | INF | 4.1 | 2.3 | 0.8 |
| Multimodal Large Language Models (Text+Image input) | | | | | | | | | | |
| GPT-5 | 3.6 | 37.6 | 51.8 | 62.5 | INF | INF | 0.0 | 3.1 | 3.8 | 53.5 |
| Gemini2.5-Flash | 5.6 | 53.1 | 37.2 | 51.1 | 0.7 | 3.1 | 40.2 | 3.4 | 3.6 | 26.1 |
| InternVL3.5-8B | 3.8 | 37.0 | 26.9 | 63.2 | INF | INF | INF | 4.7 | 4.8 | 17.5 |
| InternVL3.5-4B | 4.4 | 78.5 | 4.5 | 46.0 | INF | INF | INF | INF | INF | 0.0 |
| Qwen2.5-VL-7B | 5.6 | 31.1 | 0.2 | 47.9 | INF | INF | INF | 3.1 | 1.6 | 0.3 |
| Time Series Models | | | | | | | | | | |
| Moirai-large-311M | 10.61 | 51.7 | 100.0 | - | 0.5 | 1.4 | 100.0 | 2.0 | 2.1 | 100.0 |
| Moirai-base-91M | 10.5 | 59.8 | 100.0 | - | 0.5 | 1.4 | 100.0 | 2.0 | 2.1 | 100.0 |
| Moirai-small-14M | 77.4 | 32.9 | 100.0 | - | 0.5 | 1.5 | 100.0 | 2.0 | 2.1 | 100.0 |
| TimeMoE-base-50M | 2.9 | 38.5 | 100.0 | - | - | - | - | 2.5 | 2.7 | 100.0 |
| TimeMoE-large-200M | 3.0 | 39.0 | 100.0 | - | - | - | - | 2.6 | 2.7 | 100.0 |
| Chronos-bolt-mini-21M | 3.1 | 42.1 | 100.0 | - | - | - | - | 2.1 | 2.3 | 100.0 |
| Chronos-bolt-base-205M | 3.1 | 41.5 | 100.0 | - | - | - | - | 2.1 | 2.3 | 100.0 |
| ChaTS | - | - | - | 59.6 | - | - | - | - | - | - |
| UniTS | 3.5 | 42.0 | 100.0 | 29.5 | TMC | TMC | TMC | 3.6 | 4.0 | 96.9 |
| TimeOmni | 3.0 | 37.5 | 100.0 | 80.0 | 2.1 | 6.2 | 100.0 | 4.3 | 4.5 | 100.0 |

Table 11: Detailed results of SciTS (Economics and Neuroscience).

| Domain | Economics | Neuroscience | | | | | | | | | |
|---|---|---|---|---|---|---|---|---|---|---|---|
| Task ID | ECU03 | NEU01 | | NEU02 | | NEG03 | | | NEG04 | | |
| Metric | Acc | Acc | F1 | Acc | F1 | MAE | MAPE | SR | MAE | MAPE | SR |
| Large Language Models (Text input) | | | | | | | | | | | |
| GPT-4.1-mini | 90.4 | 50.0 | 0.0 | 23.3 | 16.1 | 49.0 | 95.2 | 96.4 | 2.7 | 22.7 | 94.4 |
| Gemini2.5-Flash | 87.8 | 50.0 | 0.0 | 9.5 | 5.8 | 8.4 | 63.5 | 99.2 | 1.9 | 13.2 | 99.2 |
| DeepSeek-V3 | 91.1 | 50.0 | 0.0 | 37.3 | 13.6 | 2.1 | 4.3 | 3.1 | 2.1 | 13.6 | 98.7 |
| Llama3-8B | 51.2 | TLS | TLS | TLS | TLS | 11.7 | 151.5 | 3.1 | 4.6 | 102.7 | 28.7 |
| Qwen3-8B | 83.5 | TLS | TLS | 16.9 | 11.2 | 9.3 | 105.8 | 26.4 | 5.2 | 63.8 | 76.1 |
| Qwen3-4B | 77.9 | TLS | TLS | 26.2 | 9.8 | 11.6 | 100.8 | 43.6 | 6.2 | 59.4 | 77.5 |
| Multimodal Large Language Models (Text+Image input) | | | | | | | | | | | |
| GPT-5 | 83.8 | 50.0 | 1.3 | 15.1 | 13.3 | 10.3 | 74.3 | 97.2 | 19.3 | 50.6 | 2.4 |
| Gemini2.5-Flash | 82.9 | 44.9 | 53.7 | 13.1 | 11.6 | INF | INF | INF | 110.7 | 97.4 | 21.2 |
| InternVL3.5-8B | 85.8 | 48.8 | 64.7 | 12.4 | 8.8 | 78.5 | 90.7 | 73.6 | 15.2 | 215.8 | 26.3 |
| InternVL3.5-4B | 86.1 | 54.6 | 20.4 | 21.2 | 7.9 | 16.7 | 189.6 | 43.9 | 328.1 | 340.4 | 28.7 |
| Qwen2.5-VL-7B | 81.9 | 50.0 | 0.0 | 0.5 | 2.4 | 20.1 | 204.8 | 18.1 | 53.5 | 749.2 | 1.9 |
| Time Series Models | | | | | | | | | | | |
| Moirai-large-311M | - | - | - | - | - | 8.1 | 59.1 | 100.0 | - | - | - |
| Moirai-base-91M | - | - | - | - | - | 7.9 | 56.5 | 100.0 | - | - | - |
| Moirai-small-14M | - | - | - | - | - | 7.9 | 56.5 | 100.0 | - | - | - |
| TimeMoE-base-50M | - | - | - | - | - | 5.9 | 66.9 | 100.0 | - | - | - |
| TimeMoE-large-200M | - | - | - | - | - | 5.8 | 70.1 | 100.0 | - | - | - |
| Chronos-bolt-mini-21M | - | - | - | - | - | 8.3 | 75.1 | 100.0 | - | - | - |
| Chronos-bolt-base-205M | - | - | - | - | - | 8.3 | 78.5 | 100.0 | - | - | - |
| ChaTS | 79.2 | 46.2 | 21.8 | 14.7 | 24.5 | - | - | - | - | - | - |
| UniTS | 27.1 | 73.4 | 63.8 | 22.7 | 13.0 | 10.7 | 95.2 | 46.4 | 66.4 | 89.3 | 100.0 |
| TimeOmni | 96.4 | 67.2 | 76.6 | 43.0 | 35.7 | 10.0 | 78.7 | 100.0 | 2.1 | 14.5 | 100.0 |

Table 12: Detailed results of SciTS (Neuroscience and Energy).

| Domain | Neuroscience | | | | Energy | | | | | |
|---|---|---|---|---|---|---|---|---|---|---|
| Task ID | NEU05 | | NEU06 | | ENG01 | | | ENG02 | | |
| Metric | Acc | F1 | Acc | F1 | MAE | MAPE | SR | MAE | MAPE | SR |
| Large Language Models (Text input) | | | | | | | | | | |
| GPT-4.1-mini | 32.6 | 25.7 | 17.1 | 12.0 | 19.6 | 218.4 | 41.9 | 4.5 | 125.0 | 1.4 |
| Gemini2.5-Flash | 30.4 | 30.9 | 19.3 | 14.0 | 8.2 | 101.7 | 56.5 | 2.5 | 72.5 | 5.9 |
| DeepSeek-V3 | 33.5 | 17.3 | 7.1 | 7.5 | 1.3 | 16.4 | 52.0 | 4.1 | 117.2 | 46.1 |
| Llama3-8B | TLS | TLS | TLS | TLS | 1.2 | 15.0 | 16.6 | 2.1 | 62.4 | 12.7 |
| Qwen3-8B | TLS | TLS | TLS | TLS | 1.0 | 13.1 | 33.3 | 2.1 | 70.2 | 18.2 |
| Qwen3-4B | TLS | TLS | TLS | TLS | 1.2 | 15.3 | 32.3 | 2.1 | 71.2 | 34.5 |
| Multimodal Large Language Models (Text+Image input) | | | | | | | | | | |
| GPT-5 | 32.0 | 25.1 | 26.2 | 16.1 | INF | INF | INF | 1.8 | 56.1 | 4.5 |
| Gemini2.5-Flash | 33.6 | 27.5 | 28.5 | 26.5 | INF | INF | INF | 3.3 | 103.9 | 7.4 |
| InternVL3.5-8B | 33.2 | 17.0 | 6.8 | 4.9 | INF | INF | INF | 2.0 | 53.0 | 0.4 |
| InternVL3.5-4B | 33.9 | 22.1 | 34.0 | 11.2 | INF | INF | INF | 2.5 | 79.2 | 0.02 |
| Qwen2.5-VL-7B | 33.4 | 16.7 | 13.7 | 4.8 | INF | INF | INF | INF | INF | INF |
| Time Series Models | | | | | | | | | | |
| Moirai-large-311M | - | - | - | - | - | - | - | 3.8 | 121.2 | 100.0 |
| Moirai-base-91M | - | - | - | - | - | - | - | 3.7 | 116.7 | 100.0 |
| Moirai-small-14M | - | - | - | - | - | - | - | 2.5 | 76.6 | 100.0 |
| TimeMoE-base-50M | - | - | - | - | - | - | - | 2.4 | 71.6 | 100.0 |
| TimeMoE-large-200M | - | - | - | - | - | - | - | 2.4 | 70.4 | 100.0 |
| Chronos-bolt-mini-21M | - | - | - | - | - | - | - | 2.5 | 72.6 | 100.0 |
| Chronos-bolt-base-205M | - | - | - | - | - | - | - | 2.5 | 73.7 | 100.0 |
| ChaTS | TMC | TMC | 10.8 | 21.9 | - | - | - | - | - | - |
| UniTS | 33.3 | 16.7 | 14.2 | 10.3 | - | - | - | 2.4 | 70.1 | 100.0 |
| TimeOmni | 50.4 | 60.4 | 73.0 | 67.8 | 1.4 | 19.3 | 100.0 | 2.2 | 68.6 | 100.0 |

Table 13: Detailed results of SciTS (Energy and Physiology).

| Domain | Energy | | | | | | | | | Physiology |
|---|---|---|---|---|---|---|---|---|---|---|
| Task ID | ENG03 | | | ENG04 | | | ENG05 | | | PHU01 |
| Metric | MAE | MAPE | SR | MAE | MAPE | SR | MAE | MAPE | SR | F1 |
| Large Language Models (Text input) | | | | | | | | | | |
| GPT-4.1-mini | 334.0 | 8.3 | 96.0 | 54.1 | 104.7 | 100.0 | 2.5 | 28.2 | 100.0 | 24.0 |
| Gemini2.5-Flash | 451.2 | 9.6 | 99.0 | 50.8 | 96.5 | 98.5 | 2.1 | 25.9 | 100.0 | 20.7 |
| DeepSeek-V3 | 294.0 | 7.7 | 98.0 | 90.2 | 61.4 | 66.7 | 2.7 | 27.8 | 100.0 | 28.9 |
| Llama3-8B | 219.5 | 8.5 | 16.0 | 0.2 | 32.8 | 19.7 | 29.4 | 82.0 | 41.5 | TLS |
| Qwen3-8B | 301.7 | 7.8 | 96.0 | 0.2 | 78.6 | 37.9 | 9.8 | 70.9 | 70.8 | TLS |
| Qwen3-4B | 291.9 | 7.5 | 99.0 | 16.1 | 90.5 | 42.4 | 24.7 | 54.6 | 72.3 | TLS |
| Multimodal Large Language Models (Text+Image input) | | | | | | | | | | |
| GPT-5 | 471.6 | 11.2 | 76.0 | 4.0 | 122.6 | 90.9 | 0.3 | 119.0 | 81.5 | 23.1 |
| Gemini2.5-Flash | 621.5 | 15.6 | 53.0 | 72.0 | 62.9 | 10.6 | 272.5 | 79.8 | 15.4 | 23.5 |
| InternVL3.5-8B | 429.6 | 42.5 | 1.0 | 43.6 | 60.1 | 93.9 | 600.8 | 6.5e3 | 56.9 | 21.1 |
| InternVL3.5-4B | 2.6e3 | 49.6 | 5.0 | 88.8 | 327.0 | 39.4 | 0.9 | 255.2 | 4.6 | 17.4 |
| Qwen2.5-VL-7B | INF | INF | 0.0 | 0.4 | 514.7 | 9.1 | INF | INF | INF | 0.2 |
| Time Series Models | | | | | | | | | | |
| Moirai-large-311M | 503.3 | 12.8 | 100.0 | 20.2 | 43.8 | 100.0 | - | - | - | - |
| Moirai-base-91M | 433.9 | 11.1 | 100.0 | 20.3 | 44.4 | 100.0 | - | - | - | - |
| Moirai-small-14M | 493.7 | 12.4 | 100.0 | 16.7 | 51.6 | 100.0 | - | - | - | - |
| TimeMoE-base-50M | 469.0 | 11.3 | 100.0 | 10.4 | 51.1 | 100.0 | - | - | - | - |
| TimeMoE-large-200M | 491.1 | 11.6 | 100.0 | 8.1 | 55.5 | 100.0 | - | - | - | - |
| Chronos-bolt-mini-21M | 514.7 | 12.4 | 100.0 | 10.4 | 38.1 | 100.0 | - | - | - | - |
| Chronos-bolt-base-205M | 516.2 | 12.0 | 100.0 | 13.2 | 38.7 | 100.0 | - | - | - | - |
| ChaTS | - | - | - | - | - | - | - | - | - | 27.5 |
| UniTS | 533.1 | 12.8 | 100.0 | 51.8 | 115.4 | 66.7 | 48.6 | 92.5 | 100.0 | 8.3 |
| TimeOmni | 300.8 | 7.4 | 100.0 | 60.0 | 111.9 | 100.0 | 6.2 | 44.1 | 100.0 | 44.5 |

Table 14: Detailed results of SciTS (Physiology).

| Domain | Physiology | | | | | | | | | |
|---|---|---|---|---|---|---|---|---|---|---|
| Task ID | PHG02 | | | PHG03 | | | PHU04 | | PHU05 | |
| Metric | MAE | MAPE | SR | MAE | MAPE | SR | Acc | F1 | Acc | F1 |
| Large Language Models (Text input) | | | | | | | | | | |
| GPT-4.1-mini | 18.3 | 1.1e3 | 94.2 | 2.3 | 20.6 | 96.0 | 57.0 | 52.7 | 89.7 | 3.0 |
| Gemini2.5-Flash | 10.1 | 110.8 | 99.0 | 0.5 | 8.9 | 98.7 | 53.2 | 64.8 | 42.0 | 19.5 |
| DeepSeek-V3 | 18.8 | 200.1 | 92.2 | 0.8 | 11.5 | 99.6 | 63.3 | 50.7 | 86.0 | 6.4 |
| Llama3-8B | 2.4 | 289.4 | 15.8 | 11.5 | 101.6 | 32.3 | 49.6 | 66.3 | 11.5 | 18.9 |
| Qwen3-8B | 8.6 | 182.7 | 37.6 | 7.9 | 59.2 | 78.6 | 50.0 | 66.7 | 11.0 | 18.8 |
| Qwen3-4B | 17.6 | 223.1 | 44.0 | 5.6 | 66.0 | 81.8 | 50.8 | 66.9 | 10.4 | 18.8 |
| Multimodal Large Language Models (Text+Image input) | | | | | | | | | | |
| GPT-5 | 16.6 | 155.3 | 97.4 | 32.2 | 182.9 | 60.5 | 45.5 | 47.8 | 56.5 | 16.9 |
| Gemini2.5-Flash | 25.0 | 185.2 | 36.9 | 296.1 | 2.3e3 | 25.1 | 42.9 | 59.0 | 16.7 | 18.3 |
| InternVL3.5-8B | 12.2 | 227.1 | 72.8 | 26.1 | 569.7 | 50.9 | 50.0 | 66.7 | 10.4 | 18.8 |
| InternVL3.5-4B | 15.2 | 855.1 | 35.7 | 125.8 | 3.1e3 | 51.8 | 49.9 | 66.6 | 10.5 | 18.8 |
| Qwen2.5-VL-7B | 15.5 | 384.3 | 21.5 | 54.7 | 950.8 | 10.1 | 43.7 | 49.6 | 87.4 | 2.5 |
| Time Series Models | | | | | | | | | | |
| Moirai-large-311M | 11.6 | 116.9 | 100.0 | - | - | - | - | - | - | - |
| Moirai-base-91M | 11.9 | 116.5 | 100.0 | - | - | - | - | - | - | - |
| Moirai-small-14M | 12.9 | 117.5 | 100.0 | - | - | - | - | - | - | - |
| TimeMoE-base-50M | 7.4 | 78.7 | 100.0 | - | - | - | - | - | - | - |
| TimeMoE-large-200M | 7.3 | 80.2 | 100.0 | - | - | - | - | - | - | - |
| Chronos-bolt-mini-21M | 11.2 | 114.8 | 100.0 | - | - | - | - | - | - | - |
| Chronos-bolt-base-205M | 10.8 | 109.3 | 100.0 | - | - | - | - | - | - | - |
| ChaTS | - | - | - | - | - | - | 51.3 | 59.5 | 10.4 | 18.7 |
| UniTS | 14.3 | 135.9 | 44.1 | 16.7 | 124.4 | 100.0 | 89.6 | 47.3 | 10.4 | 18.8 |
| TimeOmni | 15.4 | 163.0 | 100.0 | 2.1 | 20.4 | 100.0 | 90.5 | 92.7 | 38.8 | 23.0 |

Table 15: Detailed results of SciTS (Physiology and Urbanism).

| Domain | Physiology | | Urbanism | | | | | | | | |
|---|---|---|---|---|---|---|---|---|---|---|---|
| Task ID | PHU06 | | | URG01 | | | URG02 | | | URG03 | |
| Metric | Acc | F1 | MAE | MAPE | SR | MAE | MAPE | SR | MAE | MAPE | SR |
| Large Language Models (Text input) | | | | | | | | | | | |
| GPT-4.1-mini | 41.7 | 27.6 | 0.03 | 320.6 | 18.6 | 447.9 | 705.4 | 90.3 | 119.0 | 47.6 | 98.5 |
| Gemini2.5-Flash | 27.3 | 22.1 | 0.04 | 246.0 | 23.3 | 378.8 | 290.4 | 77.4 | 93.9 | 33.3 | 100.0 |
| DeepSeek-V3 | 46.3 | 22.1 | 0.04 | 350.0 | 18.6 | 686.3 | 935.6 | 50.0 | 111.6 | 37.3 | 100.0 |
| Llama3-8B | 37.8 | 12.5 | 0.03 | 282.7 | 7.0 | 1.6e3 | 2.9e3 | 4.8 | 277.2 | 300.3 | 24.6 |
| Qwen3-8B | 16.1 | 14.9 | 0.04 | 262.0 | 53.5 | 840.4 | 2.3e3 | 12.9 | 259.2 | 515.7 | 78.5 |
| Qwen3-4B | 9.0 | 4.3 | 0.03 | 158.7 | 69.8 | 571.9 | 1.1e3 | 22.6 | 289.4 | 360.2 | 76.9 |
| Multimodal Large Language Models (Text+Image input) | | | | | | | | | | | |
| GPT-5 | 36.8 | 14.1 | 0.04 | 182.1 | 58.1 | 432.5 | 704.8 | 77.4 | 610.2 | 504.5 | 55.4 |
| Gemini2.5-Flash | 31.2 | 9.3 | 0.04 | 351.9 | 16.3 | 657.4 | 1.0e3 | 40.3 | 1.4e3 | 1.4e3 | 40.0 |
| InternVL3.5-8B | 39.8 | 13.3 | 0.07 | 307.4 | 30.2 | 751.2 | 1.3e3 | 82.3 | 772.7 | 575.7 | 23.1 |
| InternVL3.5-4B | 40.8 | 9.9 | 0.05 | 253.7 | 2.3 | INF | INF | INF | 1.4e3 | 1.0e3 | 32.3 |
| Qwen2.5-VL-7B | 41.3 | 14.4 | 0.06 | 686.4 | 4.7 | INF | INF | INF | 198.1 | 137.4 | 4.6 |
| Time Series Models | | | | | | | | | | | |
| Moirai-large-311M | - | - | 0.04 | 294.7 | 100.0 | 162.8 | 189.7 | 100.0 | - | - | - |
| Moirai-base-91M | - | - | 0.04 | 274.8 | 100.0 | 160.7 | 195.5 | 100.0 | - | - | - |
| Moirai-small-14M | - | - | 0.04 | 374.9 | 100.0 | 199.8 | 287.9 | 100.0 | - | - | - |
| TimeMoE-base-50M | - | - | 0.03 | 204.4 | 100.0 | 72.5 | 55.4 | 100.0 | - | - | - |
| TimeMoE-large-200M | - | - | 0.03 | 218.4 | 100.0 | 67.2 | 53.8 | 100.0 | - | - | - |
| Chronos-bolt-mini-21M | - | - | 0.03 | 143.9 | 100.0 | 72.2 | 58.8 | 100.0 | - | - | - |
| Chronos-bolt-base-205M | - | - | 0.03 | 139.3 | 100.0 | 63.5 | 48.5 | 100.0 | - | - | - |
| ChaTS | 4.5 | 17.7 | - | - | - | - | - | - | - | - | - |
| UniTS | 5.5 | 17.4 | 0.04 | 389.7 | 100.0 | 442.5 | 1.3e3 | 69.4 | 451.1 | 628.4 | 100.0 |
| TimeOmni | 26.8 | 23.4 | 0.03 | 247.0 | 100.0 | 427.4 | 1.1e3 | 100.0 | 143.0 | 132.5 | 100.0 |

Table 16: Detailed results of SciTS (Urbanism and Manufacturing).

| Domain | Urbanism | | | | | Manufacturing | | | | | |
|---|---|---|---|---|---|---|---|---|---|---|---|
| Task ID | URU04 | | | URG05 | | MFU01 | | MFU02 | | MFU03 | |
| Metric | Acc | F1 | MAE | MAPE | SR | Acc | F1 | Acc | F1 | Acc | F1 |
| Large Language Models (Text input) | | | | | | | | | | | |
| GPT-4.1-mini | 65.6 | 64.4 | 1.2e3 | 126.6 | 100.0 | 4.5 | 3.2 | 29.3 | 34.9 | 51.2 | 56.9 |
| Gemini2.5-Flash | 53.3 | 64.6 | 1.2e3 | 98.6 | 100.0 | 20.5 | 15.1 | 27.0 | 4.6 | 50.4 | 66.8 |
| DeepSeek-V3 | 50.0 | 64.7 | 1.8e3 | 296.7 | 93.0 | 0.0 | 0.0 | 29.5 | 0.0 | 50.1 | 62.5 |
| Llama3-8B | 50.8 | 67.4 | 1.9e3 | 297.3 | 93.4 | TLS | TLS | TLS | TLS | TLS | TLS |
| Qwen3-8B | 58.2 | 66.2 | 1.5e3 | 159.4 | 89.2 | TLS | TLS | TLS | TLS | TLS | TLS |
| Qwen3-4B | 47.5 | 62.8 | 2.0e3 | 256.3 | 93.2 | TLS | TLS | TLS | TLS | TLS | TLS |
| Multimodal Large Language Models (Text+Image input) | | | | | | | | | | | |
| GPT-5 | 59.0 | 64.8 | 1.4e3 | 71.1 | 72.9 | 22.8 | 32.5 | 32.0 | 45.5 | 49.6 | 37.1 |
| Gemini2.5-Flash | 50.8 | 67.0 | 1.1e3 | 114.6 | 91.2 | 30.3 | 32.1 | 32.5 | 32.1 | 50.5 | 54.0 |
| InternVL3.5-8B | 50.8 | 67.4 | 1.5e3 | 173.9 | 99.8 | 10.0 | 0.0 | 15.3 | 21.6 | 49.8 | 37.2 |
| InternVL3.5-4B | 50.8 | 67.4 | 1.7e3 | 158.1 | 65.2 | INF | INF | INF | INF | 49.6 | 11.7 |
| Qwen2.5-VL-7B | 63.9 | 58.5 | 836.5 | 116.3 | 16.9 | 10.3 | 1.6 | 10.5 | 0.0 | 48.4 | 17.7 |
| Time Series Models | | | | | | | | | | | |
| Moirai-large-311M | - | - | 968.8 | 74.6 | 100.0 | - | - | - | - | - | - |
| Moirai-base-91M | - | - | 811.6 | 52.2 | 100.0 | - | - | - | - | - | - |
| Moirai-small-14M | - | - | 794.1 | 50.8 | 100.0 | - | - | - | - | - | - |
| TimeMoE-base-50M | - | - | 691.5 | 89.5 | 100.0 | - | - | - | - | - | - |
| TimeMoE-large-200M | - | - | 687.1 | 84.4 | 100.0 | - | - | - | - | - | - |
| Chronos-bolt-mini-21M | - | - | 759.5 | 61.1 | 100.0 | - | - | - | - | - | - |
| Chronos-bolt-base-205M | - | - | 800.6 | 70.6 | 100.0 | - | - | - | - | - | - |
| ChaTS | 48.7 | 65.4 | - | - | - | 0.0 | 0.0 | 0.0 | 0.0 | TLS | TLS |
| UniTS | 50.8 | 67.4 | TLS | TLS | TLS | 28.0 | 33.0 | 41.5 | 45.8 | 50.0 | 66.7 |
| TimeOmni | 47.5 | 64.8 | 1.7e3 | 174.0 | 100.0 | 90.0 | 90.1 | 95.8 | 94.8 | 47.8 | 61.0 |

Table 17: Detailed results of SciTS (Radar and Math).

| Domain | Radar | | | | Math | | |
|---|---|---|---|---|---|---|---|
| Task ID | RAU01 | | RAU02 | | MAG01 | | |
| Metric | Acc | F1 | Acc | F1 | MAE | MAPE | SR |
| Large Language Models (Text input) | | | | | | | |
| GPT-4.1-mini | 24.4 | 24.6 | 13.4 | 10.6 | 8.0 | 1.1e3 | 96.2 |
| Gemini2.5-Flash | 21.3 | 20.9 | 15.0 | 13.5 | 6.7 | 442.0 | 92.6 |
| DeepSeek-V3 | 27.6 | 19.4 | 11.3 | 4.2 | 5.4 | 566.0 | 95.2 |
| Llama3-8B | 16.0 | 10.8 | 13.1 | 7.5 | INF | INF | 0.0 |
| Qwen3-8B | 18.4 | 10.2 | 12.3 | 2.8 | INF | INF | 0.0 |
| Qwen3-4B | 13.6 | 8.4 | 13.1 | 8.2 | INF | INF | 0.0 |
| Multimodal Large Language Models (Text+Image input) | | | | | | | |
| GPT-5 | 26.9 | 24.3 | 10.7 | 9.1 | 4.4 | 1.4e3 | 97.9 |
| Gemini2.5-Flash | 30.8 | 31.6 | 12.5 | 11.3 | 5.2 | 510.9 | 98.1 |
| InternVL3.5-8B | 1.6 | 1.7 | 7.0 | 4.6 | INF | INF | 0.0 |
| InternVL3.5-4B | 19.7 | 6.9 | 12.5 | 2.8 | INF | INF | 0.0 |
| Qwen2.5-VL-7B | 19.9 | 6.7 | 13.4 | 5.2 | INF | INF | 0.0 |
| Time Series Models | | | | | | | |
| Moirai-large-311M | - | - | - | - | 3.2 | 360.1 | 100.0 |
| Moirai-base-91M | - | - | - | - | 3.0 | 359.9 | 100.0 |
| Moirai-small-14M | - | - | - | - | 2.0 | 311.6 | 100.0 |
| TimeMoE-base-50M | - | - | - | - | 2.4 | 573.0 | 100.0 |
| TimeMoE-large-200M | - | - | - | - | 2.4 | 1.2e3 | 100.0 |
| Chronos-bolt-mini-21M | - | - | - | - | 3.0 | 394.4 | 100.0 |
| Chronos-bolt-base-205M | - | - | - | - | 3.0 | 440.4 | 100.0 |
| ChaTS | 31.1 | 23.4 | 11.2 | 4.4 | - | - | - |
| UniTS | 18.5 | 18.4 | 12.5 | 2.8 | TLS | TLS | TLS |
| TimeOmni | 89.8 | 82.9 | 51.9 | 54.9 | 3.7 | 656.5 | 100.0 |

# G    ABLATION STUDY ON TIMEOMNI

We conducted ablations on the reprogramming module and patch-expert routing mechanism of TimeOmni to quantify the contribution of each component. Results are shown in Table 18 and Table 19.

**Ablation on the reprogramming module.** We replace the reprogramming module with a two-layer MLP using GELU activations and Dropout. This simplification leads to consistent performance degradation across most tasks. The results indicate that the reprogramming module provides stronger alignment between time-series inputs and the text LLM, improving robustness and generalization across diverse tasks.

**Ablation on the patch-expert routing module.** We replace the patch-expert routing mechanism with a fixed patch size of 1024 while keeping all other settings unchanged. Because SciTS contains very long signals, using a patch size smaller than 1024 would cause GPU out-of-memory, making 1024 the smallest feasible choice. This fixed patch size has little impact on medium-length tasks (e.g., EAU01), and can even slightly improve them, since using a single expert increases the effective training data for that patch. However, performance degrades on very long sequences (e.g., BIU03), where the model fails to capture long-range dependencies. Additionally, SciTS includes short sequences (e.g., ASU03) whose lengths are far below 1024. For these tasks, forcing a 1024-length patch collapses the entire sequence into a single patch, introducing structural inefficiencies that further hurt performance. Overall, this ablation demonstrates that patch-expert routing is essential for handling the wide range of temporal scales in scientific time series, from very short to extremely long.

Table 18: Ablation study on TimeOmni.

| Domain | Task ID | Metric | TimeOmni | w/o Reprogramming | w/o Rounter |
|--------|---------|--------|----------|-------------------|-------------|
| Astronomy | ASU01 | Acc | **69.0** | 62.3 | 58.4 |
| | | F1 | **73.5** | 70.6 | 50.4 |
| | ASG02 | MAPE | **2.8** | 10.2 | 14.3 |
| | ASU03 | Acc | **87.8** | 86.8 | 56.5 |
| | | F1 | 72.8 | **77.0** | 25.0 |
| Earth Science | EAU01 | Acc | 80.9 | 77.4 | **81.5** |
| | | F1 | 82.5 | 81.5 | **84.1** |
| | EAG02 | MAPE | **2.2** | 2.2 | 10.7 |
| Bioacoustics | BIU01 | Acc | **42.6** | 33.0 | 40.1 |
| | | F1 | **34.1** | 32.2 | 31.8 |
| | BIU02 | Acc | 83.4 | **84.2** | 83.8 |
| | | F1 | **51.2** | 44.7 | 46.7 |
| | BIU03 | Acc | **89.6** | 80.5 | 73.5 |
| | | F1 | **89.2** | 80.7 | 79.4 |
| Meteorology | MEU01 | Acc | **52.0** | 42.0 | 50.4 |
| | | F1 | **65.2** | 62.6 | 63.1 |
| | MEU02 | Acc | 48.4 | **50.8** | 48.9 |
| | | F1 | 38.8 | **44.4** | 40.6 |
| | MEG03 | MAE | **3.0** | 3.0 | 3.1 |
| | | MAPE | **37.5** | 37.7 | 38.3 |
| | MEU04 | Acc | 80.0 | 9.4 | **93.3** |
| Economics | ECG01 | MAE | 2.1 | **2.1** | 2.2 |
| | | MAPE | 6.2 | **6.1** | 6.1 |
| | ECG02 | MAE | 4.3 | 4.3 | 4.3 |
| | | MAPE | 4.5 | 4.5 | 4.5 |
| | ECU03 | Acc | **96.4** | 8.6 | 93.1 |

Table 19: Ablation study on TimeOmni (Cont.).

| Domain | Task ID | Metric | TimeOmni | w/o Reprogramming | w/o Rounter |
|---|---|---|---|---|---|
| Neuroscience | NEU01 | Acc | **67.2** | 52.3 | 64.0 |
| | | F1 | 76.6 | **80.7** | 77.7 |
| | NEU02 | Acc | 43.0 | **50.0** | 49.4 |
| | | F1 | **35.7** | 32.8 | 43.6 |
| | NEG03 | MAE | 10.0 | 9.8 | **9.6** |
| | | MAPE | **78.7** | 85.4 | 80.9 |
| | NEG04 | MAE | **2.1** | 2.4 | 7.4 |
| | | MAPE | **14.5** | 16.2 | 90.6 |
| | NEU05 | Acc | 50.4 | **50.7** | 49.6 |
| | | F1 | 60.4 | **60.8** | 59.4 |
| | NEU06 | Acc | **73.0** | 69.5 | 39.6 |
| | | F1 | **67.8** | 58.7 | 47.3 |
| Energy | ENG01 | MAE | 8.2 | 8.2 | 8.2 |
| | | MAPE | 100.0 | 100.0 | 100.0 |
| | ENG02 | MAE | 2.2 | 2.2 | **2.1** |
| | | MAPE | 68.6 | 69.2 | **66.3** |
| | ENG03 | MAE | **300.8** | 302.8 | 305.4 |
| | | MAPE | **7.4** | 7.4 | 7.5 |
| | ENG04 | MAE | 60.0 | 60.1 | **46.4** |
| | | MAPE | **111.9** | 124.7 | 116.0 |
| | ENG05 | MAE | **6.2** | 9.5 | 47.0 |
| | | MAPE | **44.1** | 42.7 | 100.8 |
| Physiology | PHU01 | F1 | 44.5 | 56.2 | **61.2** |
| | PHG02 | MAE | 15.4 | 15.2 | **14.8** |
| | | MAPE | 163.0 | 159.9 | **147.5** |
| | PHG03 | MAE | **2.1** | 2.5 | 14.5 |
| | | MAPE | **20.4** | 24.0 | 125.9 |
| | PHU04 | Acc | **90.5** | 62.9 | 60.9 |
| | | F1 | **92.7** | 83.6 | 64.4 |
| | PHU05 | Acc | 38.8 | 51.8 | **72.5** |
| | | F1 | **23.0** | 22.5 | 21.9 |
| | PHU06 | Acc | 26.8 | **54.8** | 44.8 |
| | | F1 | 23.4 | **55.5** | 46.9 |
| Urbanism | URG01 | MAE | 0.0 | 0.0 | 0.0 |
| | | MAPE | **247.0** | 295.2 | 463.0 |
| | URG02 | MAE | **427.4** | 495.8 | 455.4 |
| | | MAPE | **1057.5** | 1253.5 | 1212.7 |
| | URG03 | MAE | **143.0** | 147.9 | 436.8 |
| | | MAPE | **132.5** | 142.8 | 617.0 |
| | URU04 | Acc | 47.5 | 44.5 | **54.7** |
| | | F1 | 64.8 | 63.5 | **67.4** |
| | URG05 | MAE | 1655.8 | **1651.8** | 1662.8 |
| | | MAPE | **174.0** | 175.2 | 174.4 |
| Manufacturing | MFU01 | Acc | **90.0** | 63.8 | 29.3 |
| | | F1 | **90.1** | 71.2 | 43.2 |
| | MFU02 | Acc | **95.8** | 84.0 | 36.5 |
| | | F1 | **94.8** | 87.5 | 35.2 |
| | MFU03 | Acc | 47.8 | **48.7** | 25.3 |
| | | F1 | 61.0 | 66.6 | **66.7** |
| Radar | RAU01 | Acc | **89.8** | 87.6 | 56.3 |
| | | F1 | **82.9** | 80.6 | 45.9 |
| | RAU02 | Acc | **51.9** | 51.3 | 17.7 |
| | | F1 | **54.9** | 53.5 | 20.9 |
| Math | MAG01 | MAE | **3.7** | 3.7 | 3.9 |
| | | MAPE | 656.5 | **606.9** | 849.7 |

## H    DISCUSSION OF THE EVALUATION SETUP

TimeOmni is trained jointly across multiple domains. Its training data do not (and can hardly) cover all domains and tasks in SciTS, given the inherent sparsity of scientific time series. Tasks such as ENG04–05 and URG02–04 were evaluated in an out-of-domain setting, with detailed results provided in Appendix F. While TimeOmni may not outperform specialised time-series forecasting models on all forecasting tasks, it is able to handle tasks such as imputation and QA that those models cannot solve, and it outperforms general-purpose LLMs in both performance and success rate. TimeOmni is introduced as a proof-of-concept architecture, not as an all-purpose model or a system intended to surpass all baselines, but as an initial demonstration of how LLMs might be extended to accommodate heterogeneous and complex scientific time series.

To further address concerns about fair comparison, we additionally finetuned two open-source models, one image–text LLM and one time-series foundation model using the same training data as TimeOmni. Finetuning a text-only LLM is impractical in this setting, as converting time-series signals into text produces extremely long token sequences, making both tokenization and training prohibitively expensive. The results are shown in Tables 20-22.

We first finetuned the image-text LLM, Qwen2.5VL-7B, with LoRA of rank 64. Results show that finetuning yields performance improvement on some tasks, but the gains are limited possibly because representing time series as images compromises numerical precision. For the ENG03 task, where the model fails to follow instructions in the zero-shot setting, finetuning improves instruction-following ability, although more than half of the samples still fail to produce the required length. We further observed instability during finetuning and difficulty in maintaining balanced performance across tasks, potentially reflecting the limited capacity of image-based representations for time-series data.

We then fully finetuned TimeMoE, a time series model specialised for forecasting. Although finetuning improves its performance on some of the forecasting tasks, finetuning does not enable it to handle tasks it could not previously address, such as imputation or QA.

Table 20: Finetuning baselines on scientific time series data.

| Domain | SubTasks | Metric | TimeOmni | Qwen2.5-VL-7B (zero-shot) | Qwen2.5-VL-7B (finetuned) | TimeMoE (zero-shot) | TimeMoE (finetuned) |
|---|---|---|---|---|---|---|---|
| Astronomy | ASU01 | Acc | 69.0 | 48.4 | 48.8 | - | - |
| | | F1 | 73.5 | 2.9 | 4.7 | - | - |
| | ASG02 | MAPE | 2.8 | 6.7 | 4.9 | - | - |
| | | Success Rate | 100.0 | 1.5 | 2.4 | - | - |
| | ASU03 | Acc | 87.8 | 27.5 | 13.5 | - | - |
| | | F1 | 72.8 | 11.3 | 10.0 | - | - |
| Earth Science | EAU01 | Acc | 80.9 | 75.8 | 52.8 | - | - |
| | | F1 | 82.5 | 80.6 | 68.1 | - | - |
| | EAG02 | MAPE | 2.2 | 53.5 | 57.6 | - | - |
| | | Success Rate | 100.0 | 99.5 | 99.7 | - | - |
| Bioacoustics | BIU01 | Acc | 42.6 | 3.3 | 2.1 | - | - |
| | | F1 | 34.1 | 0.3 | 0.2 | - | - |
| | BIU02 | Acc | 83.4 | 81.0 | 82.7 | - | - |
| | | F1 | 51.2 | 22.9 | 18.1 | - | - |
| | BIU03 | Acc | 89.6 | 16.7 | 15.6 | - | - |
| | | F1 | 89.2 | 4.8 | 6.6 | - | - |
| Meteorology | MEU01 | Acc | 52.0 | 54.1 | 49.9 | - | - |
| | | F1 | 65.2 | 36.4 | 0.9 | - | - |
| | MEU02 | Acc | 48.4 | 0.0 | 73.4 | - | - |
| | | F1 | 38.8 | 0.0 | 0.0 | - | - |
| | | MAE | 3.0 | 5.5 | 2.9 | 2.9 | 2.9 |
| | MEG03 | MAPE | 37.5 | 31.1 | 23.5 | 38.5 | 36.0 |
| | | Success Rate | 100.0 | 0.2 | 0.1 | 100.0 | 100.0 |
| | MEU04 | Acc | 80.0 | 47.9 | 52.1 | - | - |

Table 21: Finetuning baselines on scientific time series data (Cont.).

| Domain | SubTasks | Metric | TimeOmni | Qwen2.5-VL-7B (zero-shot) | Qwen2.5-VL-7B (finetuned) | TimeMoE (zero-shot) | TimeMoE (finetuned) |
|---|---|---|---|---|---|---|---|
| Economics | ECG01 | MAE | 2.1 | - | - | - | - |
| | | MAPE | 6.2 | - | - | - | - |
| | | Success Rate | 100.0 | - | - | - | - |
| | ECG02 | MAE | 4.3 | 3.1 | 26.9 | 2.5 | 3.0 |
| | | MAPE | 4.5 | 1.6 | 9.3 | 2.7 | 3.3 |
| | | Success Rate | 100.0 | 0.3 | 12.0 | 100.0 | 100.0 |
| | ECU03 | Acc | 96.4 | 81.9 | 66.7 | - | - |
| Neuroscience | NEU01 | Acc | 67.2 | 50.0 | 50.0 | - | - |
| | | F1 | 76.6 | 0.0 | 0.0 | - | - |
| | NEU02 | Acc | 43.0 | 0.5 | 18.8 | - | - |
| | | F1 | 35.7 | 2.4 | 14.7 | - | - |
| | NEG03 | MAE | 10.0 | 20.1 | 42.1 | 5.9 | 6.1 |
| | | MAPE | 78.7 | 204.8 | 31.1 | 66.9 | 76.6 |
| | | Success Rate | 100.0 | 18.1 | 27.5 | 100.0 | 100.0 |
| | NEG04 | MAE | 2.1 | 53.5 | 135.1 | - | - |
| | | MAPE | 14.5 | 749.2 | 87.9 | - | - |
| | | Success Rate | 100.0 | 1.9 | 46.9 | - | - |
| | NEU05 | Acc | 50.4 | 33.4 | 33.3 | - | - |
| | | F1 | 60.4 | 16.7 | 16.7 | - | - |
| | NEU06 | Acc | 73.0 | 13.7 | 35.6 | - | - |
| | | F1 | 67.8 | 4.8 | 13.3 | - | - |
| Energy | ENG01 | MAE | 8.2 | - | - | - | - |
| | | MAPE | 100.0 | - | - | - | - |
| | | Success Rate | 100.0 | - | - | - | - |
| | ENG02 | MAE | 2.2 | - | - | 2.4 | 2.3 |
| | | MAPE | 68.6 | - | - | 71.6 | 69.4 |
| | | Success Rate | 100.0 | - | 0.0 | 100.0 | 100.0 |
| | ENG03 | MAE | 300.8 | - | 1841.8 | 469.0 | 294.7 |
| | | MAPE | 7.4 | - | 34.6 | 11.3 | 7.6 |
| | | Success Rate | 100.0 | - | 33.0 | 100.0 | 100.0 |
| | ENG04 | MAE | 60.0 | 0.4 | 703.7 | 10.4 | 17.6 |
| | | MAPE | 111.9 | 514.7 | 17.7 | 51.1 | 74.4 |
| | | Success Rate | 100.0 | 9.1 | 4.6 | 100.0 | 100.0 |
| | ENG05 | MAE | 6.2 | - | 2658.3 | - | - |
| | | MAPE | 44.1 | - | 80.3 | - | - |
| | | Success Rate | 100.0 | - | 9.2 | - | - |
| Physiology | PHU01 | F1 | 44.5 | 0.2 | 24.6 | - | - |
| | PHG02 | MAE | 15.4 | 15.5 | 41.7 | 7.4 | 7.7 |
| | | MAPE | 163.0 | 384.3 | 140.3 | 78.7 | 79.6 |
| | | Success Rate | 100.0 | 21.5 | 5.5 | 100.0 | 100.0 |
| | PHG03 | MAE | 2.1 | 54.7 | 78.5 | - | - |
| | | MAPE | 20.4 | 950.8 | 492.9 | - | - |
| | | Success Rate | 100.0 | 10.1 | 21.7 | - | - |
| | PHU04 | Acc | 90.5 | 43.7 | 50.0 | - | - |
| | | F1 | 92.7 | 49.6 | 0.0 | - | - |
| | PHU05 | Acc | 38.8 | 87.4 | 89.6 | - | - |
| | | F1 | 23.0 | 2.5 | 0.0 | - | - |
| | PHU06 | Acc | 26.8 | 41.3 | 41.4 | - | - |
| | | F1 | 23.4 | 14.4 | 15.0 | - | - |

Table 22: Finetuning baselines on scientific time series data (Cont.).

| Domain | SubTasks | Metric | TimeOmni | Qwen2.5-VL-7B (zero-shot) | Qwen2.5-VL-7B (finetuned) | TimeMoE (zero-shot) | TimeMoE (finetuned) |
|--------|----------|--------|----------|------------|-----------|------------|------------|
| Urbanism | URG01 | MAE | 0.0 | 0.1 | - | 0.0 | 0.0 |
| | | MAPE | 247.0 | 686.4 | - | 204.4 | 260.1 |
| | | Success Rate | 100.0 | 4.7 | 0.0 | 100.0 | 100.0 |
| | URG02 | MAE | 427.4 | - | 2123.0 | 72.5 | 97.6 |
| | | MAPE | 1057.5 | - | 9328.6 | 55.4 | 94.4 |
| | | Success Rate | 100.0 | - | 17.7 | 100.0 | 100.0 |
| | URG03 | MAE | 143.0 | 198.0 | 3731.3 | - | - |
| | | MAPE | 132.5 | 137.4 | 5665.8 | - | - |
| | | Success Rate | 100.0 | 4.6 | 66.2 | - | - |
| | URU04 | Acc | 47.5 | 63.9 | 48.8 | - | - |
| | | F1 | 64.8 | 58.5 | 0.0 | - | - |
| | URG05 | MAE | 1655.8 | 836.5 | 1274.2 | 691.5 | 492.2 |
| | | MAPE | 174.0 | 116.3 | 89.3 | 89.5 | 70.2 |
| | | Success Rate | 100.0 | 16.9 | 98.4 | 100.0 | 100.0 |
| Manufacturing | MFU01 | Acc | 90.0 | 10.3 | - | - | - |
| | | F1 | 90.1 | 1.6 | - | - | - |
| | MFU02 | Acc | 95.8 | 10.5 | - | - | - |
| | | F1 | 94.8 | 0.0 | - | - | - |
| | MFU03 | Acc | 47.8 | 48.4 | 50.0 | - | - |
| | | F1 | 61.0 | 17.7 | 0.0 | - | - |
| Radar | RAU01 | Acc | 89.8 | 19.9 | 19.9 | - | - |
| | | F1 | 82.9 | 6.7 | 6.7 | - | - |
| | RAU02 | Acc | 51.9 | 13.4 | 13.0 | - | - |
| | | F1 | 54.9 | 5.2 | 4.7 | - | - |
| Math | MAG01 | MAE | 3.7 | - | - | 2.4 | 2.7 |
| | | MAPE | 656.5 | - | - | 573.0 | 840.0 |
| | | Success Rate | 100.0 | - | - | 100.0 | 100.0 |

We would like to highlight that the key insight from benchmarking on SciTS is the architectural limitations of existing models—namely, their inability to support the diverse signals and task types in SciTS. As shown in Fig. 4b, the success rates and failure analyses demonstrate that these are structural issues, which cannot be resolved merely through finetuning (as further confirmed by the results above).

# I    CASE EXAMPLE

In this section, we present several case examples from SciTS. For visualisation purpose, time series are plotted as image. Note that SciTS works on raw time series.

---

**EEG signal forecasting (NEG03)**

*Predict the next time series points based on the given data.*
**Time Series**

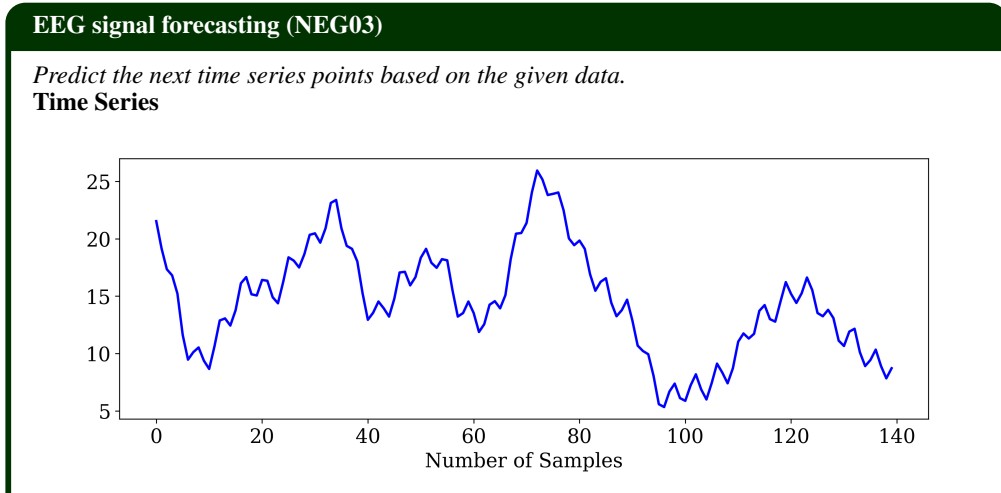

**Question:**
Now I need you to finish an EEG Recordings (ALS Patient) forecasting task. The dataset involves an ALS Patient performing self-regulation of slow cortical potentials (SCPs) to control a cursor. The sampling frequency of this dataset is 256 Hz. Please predict the next time series points given the picture above.

---

**Sleep staging classification (NEU06)**

*Determine which stage of sleep a subject is in from EEG data.*
**Time Series**

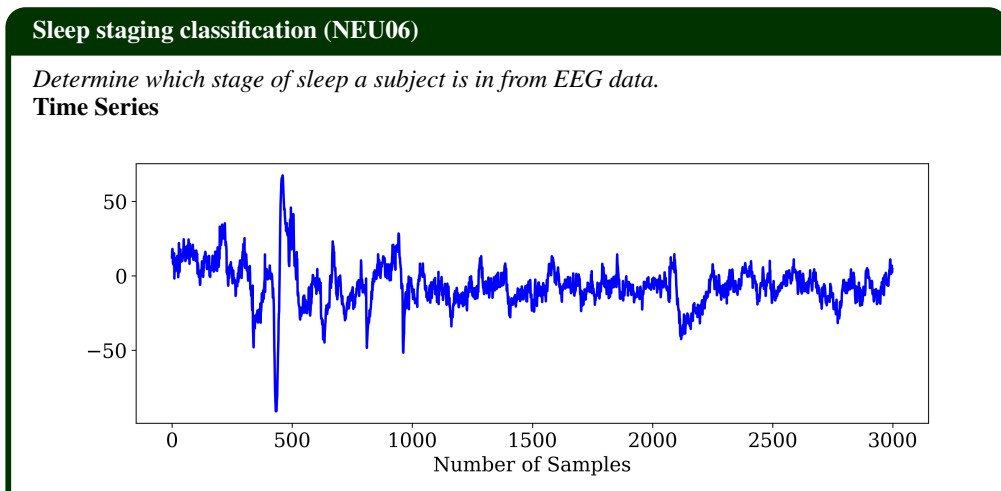

**Question:**
Now, I need your assistance to complete a sleep stage classification task. This is a single-channel EEG data from a sleeping person. Based on this data, please determine which sleep stage the person is in. It is known that the provided picture corresponds to one of the following categories:
**Choices:**
1. wakefulness stage
2. N1 stage
3. N2 stage
4. N3 stage
5. REM stage
**Answer:**
This data is classified as a wakefulness stage.

**ECG anomaly detection (PHU04)**

*Determine whether subjects exhibit abnormal conditions given their ECG data.*
**Time Series**

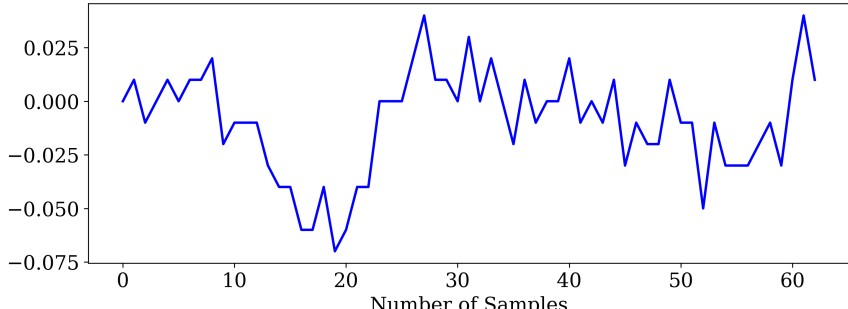

**Question:**
Please determine whether an ECG anomaly event has occurred in the provided picture. The following data represents ECG (Electrocardiogram) signals, which record the electrical activity of the heart during each heartbeat and serve as an important tool for diagnosing heart diseases. Using a signal sampled at a frequency of 500 Hz, we can determine whether abnormalities are present in the human body. If so, please indicate whether it includes Normal or Anomaly Points.
**Answer:**
Based on the given information, this time series includes a Normal Point.

**Pedestrian flow forecasting (URG02)**

*Predict the future pedestrian flow based on past data.*
**Time Series**

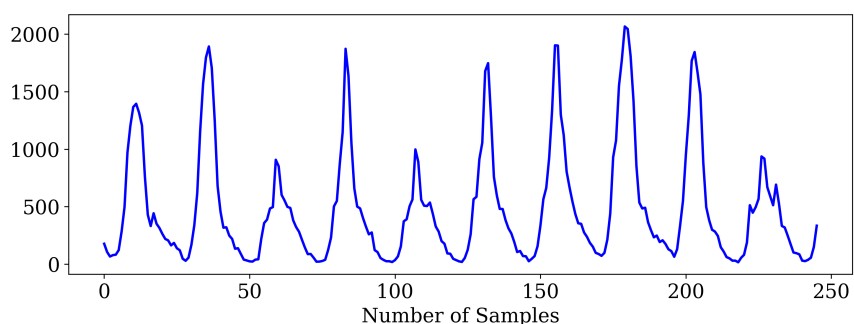

**Question:**
Now I need you to finish a Pedestrian activity forecasting task. The Melbourne Pedestrian dataset contains pedestrian count data from 10 sensor locations in the City of Melbourne, Australia, for the entire year of 2017. The dataset is designed to facilitate urban planning by analysing pedestrian activity across different times and locations, with each class representing a specific sensor placement site. The frequency is one hour. Please predict the next 9 time series points given the picture above.

**Comprehensive electricity imputation (ENG05)**

*Completes the electricity data of a certain city.*
**Time Series**

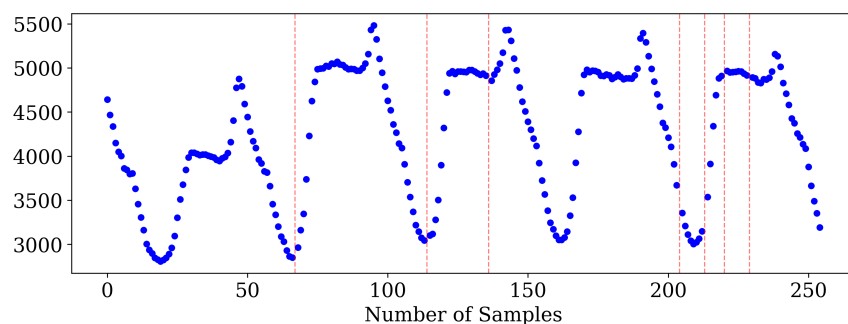

**Question:**
Now I need you to finish an Aus. Electricity Demand Imputation Task. The Aus. The Electricity Demand Dataset provides half-hourly measurements of electricity demand for Victoria, Australia, throughout the year 2014. The frequency is 30 min. Please give a full time series with 7 missing values imputed. The missing values in the picture are represented by vertical lines.

**Gravitational wave event localisation (ASU02)**

*Analyse given gravitational wave data to determine when gravitational waves occur.*
**Time Series**

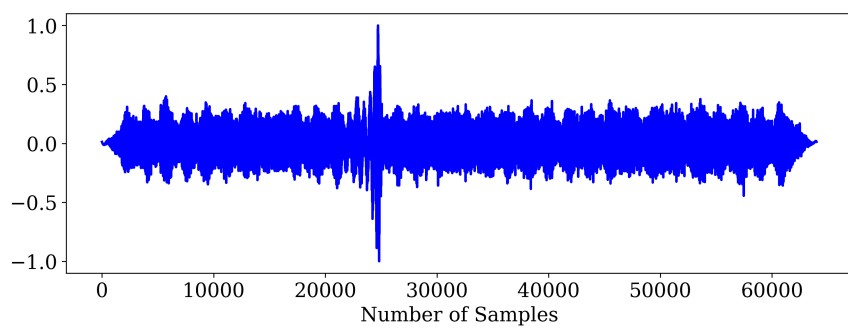

**Question:**
Please determine whether a Gravitational Wave event has occurred in the provided picture. If so, please specify the index of the starting time point of the event.
**Answer:**
A Gravitational Wave event was detected in this time-series signal. The starting time index is 24729.

### Earthquake event localisation (EAU01)

*Determine the time points of occurrence of p-wave and s-wave in the three-dimensional seismic waves.*
**Time Series**

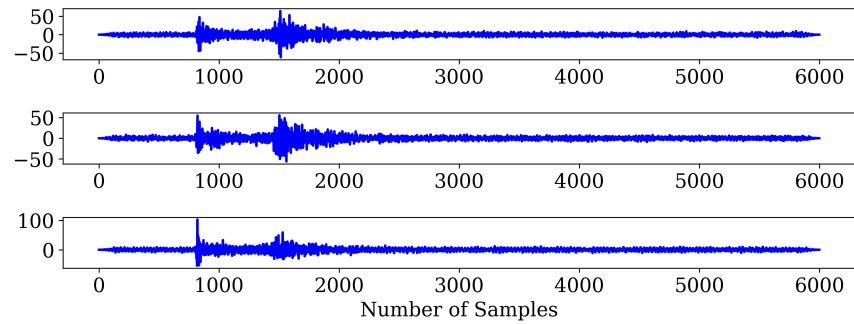

**Question:**
Please determine whether an Earthquake event has occurred in the provided picture. If so, please specify the starting time point indices of the P-wave and S-wave in the event.
**Answer:**
An Earthquake event was detected in this time-series signal. The starting time index of the P-wave is 800, and the S-wave is 1451.

### Stock MCQ (ECU03)

*Analyse the given time series data to solve the multi-choice question.*
**Time Series**

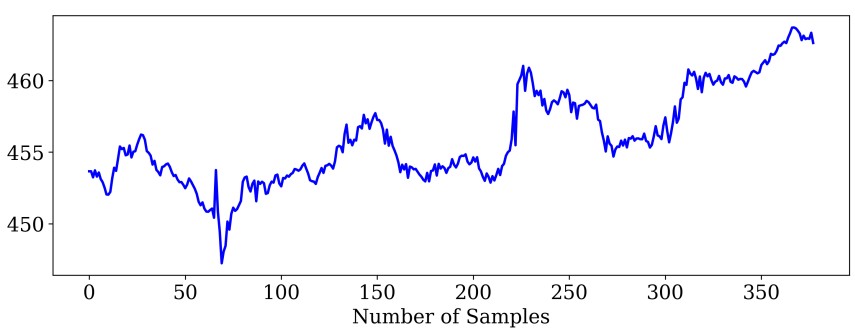

**Question:**
Now I need you to finish a Stock QA task. This data is used to analyse which of the four options, A, B, C, and D is correct. Please answer the question based on the given picture.
**Choices:**
A. The consistent increase in stock price post-news indicates a positive market reception to the investment recommendations provided in the financial report, suggesting bullish sentiment among investors.
B. The price movement shows volatility without any clear trend, indicating investor uncertainty regarding UNHŽ019s future performance despite the positive news.
C. The stock price fluctuations in the days following the news publication demonstrate a weak correlation with changes in earnings estimates, implying that investors were not influenced by the market data provided.
D. The stock price decreased significantly after the news announcement, suggesting that the market reacted negatively to the suggested investment in UNH.
**Answer:**
Option A is correct.

