# OpenReview forum: "SciTS: Scientific Time Series Understanding and Generation with LLMs"
_ICLR.cc/2026/Conference — ICLR 2026 Poster_

### Official Review · Reviewer_5o2d · 2025-10-24

**Soundness:** 3
**Presentation:** 3
**Contribution:** 2
**Rating:** 6
**Confidence:** 3

**Summary:**

This paper introduces SciTS, a large-scale benchmark for evaluating the ability of LLMs to understand and generate scientific time series across 12 domains, 43 tasks, and over 52k instances. It further proposes TimeOmni, a modular framework that integrates explicit temporal encoding, router-based patch experts, and patch reprogramming for unifying diverse scientific signals within LLM architectures.

**Strengths:**

1. SciTS fills a gap by offering a unified high-quality evaluation suite for scientific time series
2. The benchmark includes 17 baselines across modalities with consistent metrics, which is good for cross-domain generalization and model behavior.

**Weaknesses:**

1. TimeOmni uses fine-tuning while others are evaluated purely zero-shot, which would impact strict comparability. An ablation without fine-tuning would strengthen the claim.
2. No ablation for key modules, like router, patch reprogramming, and expert families.
3. It remains unclear how TimeOmni handles very long or high-frequency sequences beyond benchmark scales.
4. Benchmark provenance and overlap with LLM pre-training corpora are not fully documented.

**Questions:**

1. How is normalization handled across domains with very different frequency/time scales?
2. Were all non-TimeOmni baselines strictly zero-shot? If so, could you include a variant of TimeOmni under the same condition?
3. How stable and interpretable is the router-based patch expert mechanism during inference?
4. Do you plan to release detailed data provenance and licensing information to ensure benchmark sustainability?
5. Could the framework extend to spatio-temporal or higher-dimensional scientific data (e.g., radar or 3D simulations)?

---

> ### Author Response · Authors · 2025-11-19
> **Response to Reviewer 5o2d (1/2)**
>
> We thank reviewer for the valuable comments and we respond to the questions below.
>
> **[Regarding evaluation setup]**
>
>   Since scientific time series are sparse and hard to collect, the training data of TimeOmni in fact do not (and can hardly) cover all domains and tasks in SciTS. Tasks such as ENG04–05 and URG02–04 were evaluated in **zero-shot setting**, with detailed results provided in Appendix G. While TimeOmni may not outperform specialised time-series forecasting models on all forecasting tasks, it is able to handle tasks such as imputation and QA that those models cannot solve, and it outperforms general-purpose LLMs in both performance and success rate.
>
> Having said that, we would like to emphasise that the primary insight obtained from benchmarking on SciTS lies in the architectural limitations of existing models, namely their inability to accommodate the diverse data modalities and task types present in scientific time series (see Fig. 4). These limitations are structural in nature and cannot be resolved through finetuning alone. Our goal is to bring this challenge to the community’s attention and motivate further exploration in this direction. TimeOmni is therefore introduced as **a proof-of-concept framework**, not as an all-purpose model or a system designed to surpass all baselines, but as an illustrative step toward enabling LLMs to process heterogeneous and complex scientific time series.
>
> To further address concerns about fair comparison, we additionally finetuned two open-source models, one image–text LLM and one time-series foundation model, on a subset of benchmark tasks, within our limited time and computational resources. Finetuning a text-only LLM is impractical in this setting, as converting time-series signals into text produces extremely long token sequences, making both tokenization and training prohibitively expensive. The results, reported below, include success rate (SR), mean absolute error (MAE), and mean absolute percentage error (MAPE).
>
> | Task  | Metric | Qwen2.5-VL-7B (zero-shot) | Qwen2.5-VL-7B (finetune) | TimeMoE-base (zero-shot) | TimeMoE-base (finetune) | TimeOmni |
> |-------|---------|----------------------------|----------------------------|---------------------------|---------------------------|-----------|
> | **ASU01** | Acc $\uparrow$| 48.4 | 49.9 | – | – | 69.0 |
> |       | F1 $\uparrow$ | 2.9 | 8.6 | – | – | 73.5 |
> | **ASG02** | MAPE $\downarrow$ | 6.7 | 5.2 | – | – | 2.8 |
> |       | SR $\uparrow$| 1.5 | 4.5 | – | – | 100.0 |
> | **ASU03** | Acc $\uparrow$| 27.5 | 30.1 | – | – | 87.8 |
> |       | F1 $\uparrow$| 11.3 | 13.5 | – | – | 72.8 |
> | **EAU01** | Acc $\uparrow$| 75.8 | 75.8 | – | – | 80.9 |
> |       | F1 $\uparrow$| 80.6 | 80.6 | – | – | 82.5 |
> | **EAG02** | MAPE $\downarrow$| 53.5 | 54.3 | – | – | 2.2 |
> |       | SR $\uparrow$| 99.5 | 99.7 | – | – | 100.0 |
> | **BIU03** | Acc $\uparrow$| 16.7 | 15.9 | – | – | 89.6 |
> |       | F1 $\uparrow$| 4.8 | 8.0 | – | – | 89.2 |
> | **ENG03** | MAE $\downarrow$| - | 579.9 | 469.0 | 417.6 | 300.8 |
> |        | MAPE $\downarrow$| - | 13.3 | 11.3 | 10.5 | 7.4 |
> |        | SR $\uparrow$| 0 | 46.0 | 100.0 | 100.0 | 100.0 |
>
> 1. We first finetune the image-text LLM, Qwen2.5VL-7B, with LoRA. Results show that finetuning yields performance improvement on some tasks, but the gains are limited possibly because representing time series as images compromises numerical precision. For the ENG03 task, where the model fails to follow instructions in the zero-shot setting, finetuning improves instruction-following ability, although more than half of the samples still fail to produce the required length.
> 2. We then fully finetune TimeMoE, a time series model specialised for forecasting. Although finetuning improves its performance on forecasting tasks, finetuning does not enable it to handle tasks it could not previously address, such as imputation or QA.
>
>
> **[Regarding very long or high-frequency sequences]**
>
> SciTS already includes very long (10M points) and high-frequency (10 MHz) sequences. TimeOmni's flexible dynamic patching router can be adjusted for even longer sequences, such as adjusting or adding extra experts of larger patch sizes.
>
>
> **[Benchmark provenance and overlap with LLM pre-training corpora]**
>
> The data sources used to construct SciTS are documented in Appendix D.2, and we will release the benchmark upon acceptance of the paper. As for LLM pretraining corpora, these are generally not publicly disclosed, so we are unable to fully verify potential overlap. However, for text-only LLMs or image-text LLMs, their training data typically do not include scientific time series of these types. For open-source time series models (i.e., TimeMoE, Moirai, Chronos, UniTS, ChaTS), we have confirmed that there's no overlap between SciTS and their pretraining data.
>
> *(The rest of your comments will be addressed in Response 2/2).*

---

> ### Author Response · Authors · 2025-11-19
> **Response to Reviewer 5o2d (2/2)**
>
> **[Regarding ablation studies]**
>
> We provide ablation results below to quantify the contribution of each component. Again, we report the results on a subset of benchmark tasks due to time and resource constraints.
>
> | Task   | Metric | TimeOmni | TimeOmni w/o Reprogramming | TimeOmni w/o Router |
> |--------|--------|----------|--------------------------------------|------------------------------------|
> | **ASU01** | Acc $\uparrow$ | 69.0  | 62.3  | 58.4 |
> |      | F1  $\uparrow$ | 73.5  | 70.6 | 50.4 |
> | **ASU03** | Acc $\uparrow$ | 87.8  | 86.8 | 56.5 |
> |      | F1  $\uparrow$ | 72.8  | 77.0 | 25.0 |
> | **EAU01** | Acc $\uparrow$| 80.9  | 77.4 | 81.5 |
> |      | F1  $\uparrow$ | 82.5  | 81.5  | 84.1 |
> | **BIU03** | Acc $\uparrow$ | 89.6  | 80.5 | 73.5 |
> |      | F1 $\uparrow$ | 89.2  | 80.7 | 79.4 |
> | **ASG02** | MAPE $\downarrow$ | 2.8   | 10.2 | 14.3 |
> | **EAG02** | MAPE $\downarrow$ | 2.2   | 2.2   | 10.7 |
> | **ENG03** | MAPE $\downarrow$  | 7.4   | 7.4  | 7.5 |
>
>
> 1. **Ablation on the reprogramming module:** We replace the reprogramming module with a two-layer MLP using GELU activations and Dropout. This simplification leads to consistent performance degradation across most tasks. The results indicate that the reprogramming module provides stronger alignment between time-series inputs and the text LLM, improving robustness and generalization across diverse tasks.
> 2. **Ablation on the patch-expert routing module:** We replace the patch-expert routing mechanism with a fixed patch size of 1024 while keeping all other settings unchanged. Because SciTS contains very long signals, using a patch size smaller than 1024 would cause GPU out-of-memory, making 1024 the smallest feasible choice. This fixed patch size has little impact on medium-length tasks (e.g., EAU01), and can even slightly improve them, since using a single expert increases the effective training data for that patch. However, performance degrades on very long sequences (e.g., BIU03), where the model fails to capture long-range dependencies. Additionally, SciTS includes short sequences (e.g., ASU03) whose lengths are far below 1024. For these tasks, forcing a 1024-length patch collapses the entire sequence into a single patch, introducing structural inefficiencies that further hurt performance. Overall, this ablation demonstrates that patch-expert routing is essential for handling the wide range of temporal scales in scientific time series, from very short to extremely long.
>
> **[Regarding normalization]**
>
> We perform normalisation at instance-level. Sepcifically, we normalise each data sample individually, with each channel normalised separately. Although the data from different domains have very different frequency/time scales, this does not affect the normalisation of each individual sample.
>
>
> **[Regarding stability of the patch expert mechanism]**
>
> The patch–expert router is rule-based and is therefore stable and interpretable during inference. To ensure that the output length is approximately $L$ ($L=100$ in the paper) regardless of the input signal length $T$, the router simply selects the patch experts with the largest patch size based on the condition: $L_k < \frac{T}{L}$ where $L_1, L_2, ..., L_N$ are the patch lengths of the Patch Experts.
>
> **[Extension of the framework to spatio-temporal or higher-dimensional  data]**
>
> Yes. TimeOmni is designed to support flexible, multi-dimensional data, as demonstrated on SciTS, and can be extended to handle spatio-temporal or higher-dimensional scientific data. Code will be release upon acceptance of the paper.
>
>
>
>
> We thank the reviewer again for the valuable comments. We would like to mention that this paper is submitted to the *"Datasets and Benchmarks"* track, with the SciTS benchmark as its intended primary contribution: the first comprehensive benchmark for scientific time series generation and understanding. We sincerely appreciate the reviewers’ recognition of this contribution. Our extensive evaluation reveals that existing architectures fail to support all signal and task types in SciTS. Our goal is to raise attention to this problem and encourage further research. TimeOmni is then presented as a proof-of-concept demonstration, not as a definitive solution, but as an illustrative step toward LLMs capable of handling complex and heterogeneous scientific signals. We hope our responses fully resolve your concerns.

---

### Official Review · Reviewer_pjb9 · 2025-10-30

**Soundness:** 3
**Presentation:** 3
**Contribution:** 3
**Rating:** 4
**Confidence:** 4

**Summary:**

The paper collects a comprehensive set of scientific related multimodal time series datasets, spanning various tasks including forecasting, classification, anomaly detection, etc. It benchmarks text-only LLMs, multimodal LLMs, and unified time series models and addresses the limitations of existing approaches by its proposed TimeOmni. TimeOmni fine-tunes an LLM by aligning time series representations and text representations, with separate output heads for different tasks. Extensive experiments across the benchmark show that TimeOmni achieves strong performance relative to all three categories of existing methods.

**Strengths:**

1. The paper contributes a large-scale multimodal time series benchmark covering diverse tasks and domains.

2. The paper conducted broad evaluation comparing both text-only LLMs, multimodal LLMs, and unified time series models.

3. The authors propose a new model that aligns time series and text, and show through extensive experiments that the model outperforms all three types of existing methods.

**Weaknesses:**

1. Is TimeOmni trained and evaluated per domain, or is it trained jointly across all domains and tasks? The compared models, even for those open-source models with the same scale, are evaluated in a zero-shot setting, so it seems unfair for those models as TimeOmni has been fine-tuned on the target domains. I may have missed this part, but have the authors tried any out-of-domain testing?

2. The current design does not support dynamic-length generation, which limits flexibility and scalability compared to sequence decoders that can have variable-length outputs.

3. Apart from per-domain performance, are there any results on per-task performance? For example, how does TimeOmni compare with current time series models on the forecasting task? Practitioners may not care about having a general model that can handle all the tasks but more on the performance for a specific task of their interest.

**Questions:**

1. Have the authors explored RL training after supervised fine-tuning?

2. For multivariate time series with dimensions up to 58 in Neuroscience, would the current representations make the input length very long?

3. How to compute the scores of a task if a part of samples fail this task? For example, did the authors impute zeros to compute MAPE for those failed samples?

---

> ### Author Response · Authors · 2025-11-19
> **Response to Reviewer pjb9 (1/2)**
>
> We thank the reviewer for the valuable comments and for acknowledging the soundness and contribution of our work. We respond to each question below.
>
> **[Regarding test setting]**
>
> TimeOmni is trained jointly across multiple domains, but its training data do not (and can hardly) cover all domains and tasks in SciTS, given the inherent sparsity of scientific time series. Tasks such as ENG04–05 and URG02–04 are evaluated in an **out-of-domain** setting, with detailed results provided in Appendix G. While TimeOmni may not outperform specialised time-series forecasting models on all forecasting tasks, it is able to handle tasks such as imputation and QA that those models cannot solve, and it outperforms general-purpose LLMs in both performance and success rate.
>
> We would like to highlight that the purpose of benchmarking on SciTS is to reveal the architectural limitations of existing models, specifically, their inability to handle the diverse data modalities and task types encompassed by scientific time series (as illustrated in Fig. 4). These shortcomings are structural rather than data-related, and therefore cannot be resolved through finetuning alone. In this context, TimeOmni is introduced as **a proof-of-concept architecture**, not as an all-purpose model or a system intended to surpass all baselines, but as an initial demonstration of how LLMs might be extended to accommodate heterogeneous and complex scientific time series.
>
> To further address the concern regarding fair comparison, we finetuned two open-source models, one image-text LLM and one time-series foundation model, on a number of benchmark tasks due to limited time and resources. Finetuning a text-only LLM becomes impractical because time-series signals translate into very long sequences, making both tokenisation and training computationally expensive. Results are shown below, where SR, MAE, and MAPE stand for success rate, mean absolute error and mean absolute percentage error, separately.
>
>
> | Task  | Metric | Qwen2.5-VL-7B (zero-shot) | Qwen2.5-VL-7B (finetune) | TimeMoE-base (zero-shot) | TimeMoE-base (finetune) | TimeOmni |
> |-------|---------|----------------------------|----------------------------|---------------------------|---------------------------|-----------|
> | **ASU01** | Acc $\uparrow$ | 48.4 | 49.9 | – | – | 69.0 |
> |       | F1 $\uparrow$ | 2.9 | 8.6 | – | – | 73.5 |
> | **ASG02** | MAPE $\downarrow$| 6.7 | 5.2 | – | – | 2.8 |
> |       | SR $\uparrow$ | 1.5 | 4.5 | – | – | 100.0 |
> | **ASU03** | Acc $\uparrow$ | 27.5 | 30.1 | – | – | 87.8 |
> |       | F1 $\uparrow$ | 11.3 | 13.5 | – | – | 72.8 |
> | **EAU01** | Acc $\uparrow$ | 75.8 | 75.8 | – | – | 80.9 |
> |       | F1 $\uparrow$ | 80.6 | 80.6 | – | – | 82.5 |
> | **EAG02** | MAPE $\downarrow$| 53.5 | 54.3 | – | – | 2.2 |
> |       | SR $\uparrow$ | 99.5 | 99.7 | – | – | 100.0 |
> | **BIU03** | Acc $\uparrow$ | 16.7 | 15.9 | – | – | 89.6 |
> |       | F1 $\uparrow$ | 4.8 | 8.0 | – | – | 89.2 |
> | **ENG03** | MAE $\downarrow$ | - | 579.9 | 469.0 | 417.6 | 300.8 |
> |        | MAPE $\downarrow$ | - | 13.3 | 11.3 | 10.5 | 7.4 |
> |        | SR $\uparrow$ | 0 | 46.0 | 100.0 | 100.0 | 100.0 |
>
> 1. We first finetune the image-text LLM, Qwen2.5VL-7B, with LoRA. Results show that finetuning yields performance improvement on some tasks, but the gains are limited possibly because representing time series as images compromises numerical precision. For the ENG03 task, where the model fails to follow instructions in the zero-shot setting, finetuning improves instruction-following ability, although more than half of the samples still fail to produce the required length.
> 2. We then fully finetune TimeMoE, a time series model specialised for forecasting. Although finetuning improves its performance on forecasting tasks, finetuning does not enable it to handle tasks it could not previously address, such as imputation or QA.
>
>
> **[Regarding dynamic-length generation]**
>
> TimeOmni does support dynamic-length, as verified on SciTS which involves tasks with diverse generation length. We set up multiple regression heads to output time series of different lengths. Specifically, we configured regression heads for predicting lengths of 1, 2, 4, ..., 1024. For a given target length $L$, the model selects the smallest predicted length $p_l$ such that $p_l \geq L$, and then truncates the first $L$ elements to produce the final prediction. This enables dynamic-length generation and can be extended to longer sequences. While less scalable than auto-regressive generation methods, this non-autoregressive approach is more efficient and avoids error accumulation from cascaded generation.
>
> *(The rest of your comments will be addressed in Response 2/2).*

---

> ### Author Response · Authors · 2025-11-19
> **Response to Reviewer pjb9 (2/2)**
>
> **[Regarding results on per-task performance]**
>
> Yes, full per-task results are provided in Appendix G. We note that there is an inherent trade-off between specialised models trained for specific tasks and general models capable of handling diverse tasks. The primary goal of the SciTS benchmark is to evaluate and highlight the ability of current general-purpose LLMs to process a wide range of scientific signals and tasks, rather than to optimise performance for a single task.
>
> **[Regarding RL training]**
>
> We would like to thank the reviewer for this suggestion. Since tasks in SciTS have single ground-truth answers, supervised fine-tuning is a suitable and sufficient training approach at this stage. The primary goal of this paper is to bring attention to an important but largely underexplored challenge in scientific AI: assessing the capability of LLMs to understand and generate scientific time series, and encourage further research in this field. TimeOmni then serves as an illustrative structural framework, providing a preliminary approach to inspire extensions and improvements, including potential RL-based training or other future directions.
>
>
> **[Regarding multivariate time series]**
>
> Higher input dimensionality does increase the raw sequence length. To address this, we use the Patch Experts Router, which dynamically selects appropriate patch sizes so that the post-patching sequence length stays roughly fixed (≈100 tokens in our experiments), regardless of the original input dimensionality or sequence length.
>
> **[Regarding metric computation under task failure]**
>
> As stated in Section 5 (line 296-301) and in caption of Table 4, we report success-rate weighted MAPE (swMAPE), where success rate is defined as the proportion of successfully processed samples relative to the total number of samples. swMAPE is then defined as MAPE divided by the success rate, providing an overall performance measure for generation tasks that penalises failures.
>
>
> We thank the reviewer again for the valuable feedback. We would like to clarify that this paper is submitted to the **"Datasets and Benchmarks"** track, with the SciTS benchmark as its primary contribution which presents the limitation of existing architectures in supporting diverse and complex scientific signals. We sincerely appreciate the reviewers’ recognition of this contribution. TimeOmni then serves as an illustrative extension towards exploring how models might handle complex and heterogeneous scientific signals. Our intention is to spark broader attention and to encourage further development of more powerful and comprehensive approaches to address this challenging research problem. We hope our responses fully address your concern and we are more than happy to elaborate further.

---

### Official Review · Reviewer_4z7J · 2025-10-31

**Soundness:** 2
**Presentation:** 3
**Contribution:** 2
**Rating:** 2
**Confidence:** 4

**Summary:**

The paper titled “SciTS: Scientific Time Series Understanding and Generation with LLMs”  introduces SciTS, a new benchmark for scientific time series understanding and generation, spanning a number of disciplines, tasks and data with diverse signal characteristics. The authors benchmark 17 models, revealing that general-purpose LLMs often show better generalization than specialized time series models, but their performance is limited when time series are converted to text (due to long sequences) or images (due to loss of precision). To address these challenges, the paper proposes TimeOmni, an LLM-based framework that explicitly models temporal dynamics and supports both time series understanding and generation, achieving the top rank on the challenging SciTS benchmark.

**Strengths:**

The benchmark provided is pretty exhaustive for scientific time series. It consists of 52,056 instances spanning 43 domain-specific tasks across 12 scientific disciplines. It has has diversity types of data including both univariate and multivariate. It also includes various task types such as anomaly detection, classification, multiple-choice question answering (MCQ), event localization, forecasting, imputation, and synthesis

The proposed TimeOmni demonstrates its effectiveness by achieving the highest overall ranking on the challenging SciTS benchmark, underscoring the advantage of its approach.

**Weaknesses:**

The paper is primarily a benchmark contribution (SciTS), with the proposed TimeOmni methodology being a secondary, and arguably incremental, focus. While the benchmark is vast, this heavy reliance on data creation means the paper's core scientific novelty may be perceived as low, as the methodological innovation is not groundbreaking.

The creation of the SciTS benchmark largely involves the collection and curation of existing open-source datasets and data from scientific domain websites, alongside some numerical simulation methods. While the combination, annotation, and unifying under a prompt-based format are novel contributions , the underlying raw time series signals themselves are largely drawn from pre-existing sources.

The proposed TimeOmni framework is an adaptation of existing LLM componentsFor instance, it uses a Patch Reprogramming module which is a concept previously presented in Time-LLM (Jin et al., 2024),  the router and patch family are basically selective resizing.

The title, "SCITS: SCIENTIFIC TIME SERIES UNDERSTANDING AND GENERATION WITH LLMS," is misleading. The title should use the word benchmark because that is what it is. Potential suggestion:
"SCITS: A NEW BENCHMARK FOR SCIENTIFIC TIME SERIES UNDERSTANDING”

**Questions:**

na

---

> ### Author Response · Authors · 2025-11-19
> **Response to Reviewer 4z7J**
>
> We thank the reviewer for the valuable comments and respond to the questions as follows.
>
> **[Primary contribution]**
>
> We would like to first clarify that the paper was submitted to the **"Datasets and Benchmarks"** Track, and the scientific time-series benchmark SciTS is the primary intended contribution.
>
> **[The scientific novelty of the SciTS benchmark]**
>
> The goal of SciTS is not simply to aggregate data. Rather, it raises a new and critical research question: *Can multimodal LLMs process an exceptionally diverse range of scientific time series within a unified framework?* This represents a fundamentally important yet largely underexplored direction in scientific AI research.
>
> Scientific time series are ubiquitous across disciplines, but they are exceptionally heterogeneous in length, dimensionality, physical constraints, and domain-specific semantics, making them fundamentally challenging to model. Benchmark results demonstrate that existing textual and visual LLMs are structurally incapable of solving these tasks. Their systematic failures reveal previously unaddressed scientific challenges that current architectures were not designed to handle, underscoring the scientific novelty and necessity of the benchmark. Enabling LLMs to effectively process such signals is crucial for building artificial general intelligence (AGI) systems that can meaningfully support scientific analysis and discovery.
>
> Unlike images or text, which can be readily captured or generated, scientific time series such as gravitational waves or seismic measurements cannot be freely “created”. Although benchmark construction does not require generating new data from scratch, scientific datasets are fundamentally constrained by what can be measured in the real world. Our work is the first to collect and unify such a broad diversity of scientific time-series domains and tasks. Even aggregating existing sources and integrating them into a coherent, comprehensive benchmark demands substantial scientific and technical effort, far beyond a straightforward data-collection exercise.
>
> **[TimeOmni framework]**
>
> As discussed in the paper, existing LLM architectures, whether serialising temporal signals as text or rasterising them as images, are structurally unable to address the heterogeneous tasks in SciTS. Our intention is to draw attention to this critical challenge and to motivate further research in this direction. We introduce TimeOmni as an initial step: a minimal-modification approach designed to enable LLMs to process diverse scientific time series without extensive architectural changes (as also noted by the reviewer). It should be viewed as **a proof-of-concept demonstration** rather than a fully developed, all-purpose model, illustrating one possible path toward more capable time-series–aware LLMs.
>
> Specifically, to achieve this, we made following extensions beyond TimeLLM framework:
> 1. We introduced multiple Patch Experts with different patch sizes, and used a Router to select the appropriate Patch Expert for input time series of varying lengths. This is critical to handle diverse scientific time series spanning lengths from $10^0$ to $10^7$. Otherwise, long sequences with a small patch would leads to GPU out-of-memory or failure of capturing too long-range dependencies, while short sequences with a large patch could result in the sequence collapsing to a single patch, introducing inherent design inefficiencies and degrading performance.
> 2. TimeLLM is limited to time series prediction tasks. In contrast, by leveraging the text token output from the LLM, TimeOmni is capable of simultaneously handling both understanding (generating text tokens) and generation (producing time series values) tasks.
> 3. For the generation task, TimeOmni further incorporated multiple regression heads to output time series of different lengths.
>
> We thank the reviewer again for the valuable feedback and would like to emphasise that the primary contribution of this work is the SciTS benchmark: the first comprehensive evaluation suite spanning diverse scientific time-series domains and task types. SciTS highlights an important yet largely under-explored challenge in scientific AI: whether and how LLMs can understand, process, and generate scientific time series. Our aim is to foreground this problem and stimulate further research in this direction. TimeOmni is offered as a preliminary and illustrative approach rather than a definitive solution. It demonstrates one feasible pathway for extending LLMs to operate over heterogeneous scientific time series, serving as an example of what future, more capable architectures might build upon. We hope our responses fully address you concern and we are more than happy to elaborate further.

---

### Official Review · Reviewer_P26w · 2025-11-02

**Soundness:** 3
**Presentation:** 3
**Contribution:** 3
**Rating:** 6
**Confidence:** 3

**Summary:**

This paper introduces SCITS (Scientific Time Series Understanding and Generation) — a comprehensive benchmark encompassing 12 scientific disciplines, 43 tasks, and over 50,000 instances — designed to evaluate the capability of large language models (LLMs) in processing scientific time series data. The authors observe that existing multimodal LLMs either encode time series as text (leading to excessive sequence lengths) or as images (losing numerical precision). To bridge this gap, the paper proposes TimeOmni, an LLM-compatible framework that explicitly models temporal dynamics through patch experts and adaptive routing, enabling both understanding and generation of time series while remaining compatible with general LLM training. Extensive benchmarking across 17 models—including GPT-5, Gemini-2.5, Qwen3, and Moirai—demonstrates that TimeOmni consistently outperforms both general LLMs and specialised time-series models across most scientific domains.

**Strengths:**

1. The introduction of SCITS represents a major step forward for LLM-based scientific data understanding. The benchmark’s diversity—covering astronomy, neuroscience, meteorology, physiology, and more—significantly broadens the scope beyond traditional forecasting or anomaly detection benchmarks.
2. The dual emphasis on understanding (classification, QA, anomaly detection) and generation (forecasting, imputation, synthesis) is novel, establishing SCITS as arguably the first unified evaluation for both reasoning and signal generation.
3. The manuscript is clearly written, logically structured, and well-illustrated.
4. TimeOmni not only achieves full task coverage but also top-1 average ranking across nearly all disciplines, demonstrating both robustness and generality.

**Weaknesses:**

1. The paper briefly mentions recent works like Time-LLM and ChatTS, but deeper discussion of conceptual differences and computational efficiency trade-offs would help position TimeOmni more clearly in the landscape.
2. The paper evaluates models in a zero-shot setting only. While this fairly tests generalisation, it leaves open the question of how much performance could improve with lightweight adaptation, e.g., LoRA or task-specific finetuning.
3. Although the patch-expert routing and reprogramming modules are key innovations, the paper does not include ablations isolating their contributions. Quantifying improvements from each would strengthen the architectural claims.

**Questions:**

1. Given that the router dynamically selects patch experts, what is the computational overhead compared to standard LLM inference? Is the training cost comparable to multimodal extensions (e.g., audio or vision encoders)?
2. How does TimeOmni perform under few-shot or domain-specific fine-tuning? Could small-scale adaptation bridge the performance gap between open-source and closed-source LLMs?
3. Could the authors provide quantitative ablation results on the router and patch-reprogramming components to confirm their necessity?

---

> ### Author Response · Authors · 2025-11-19
> **Response to Reviewer P26w (1/3)**
>
> We thank the reviewer for the valuable comments and for acknowledging the soundness and contribution of the paper. Our responses are as follows:
>
> **[Regarding Recent Works]**
>
> Due to page limit, we provided a detailed discussion of the related work in Appendix B.1. We further provide a discussion of conceptual differences and computational efficiency trade-offs below and will add it to the final version:
>
> *"Existing approaches that adapt multimodal large language models (MLLMs) to time series processing face  limitations. Directly feeding raw time series signals into text-based LLMs results in excessively long token sequences, leading to high memory consumption and poor computational efficiency due to the quadratic complexity of self-attention mechanisms. Alternatively, treating time series as images and inputting them into vision-language models introduces limitations suffers from loss of numerical precision, which degrades model performance, especially for fine-grained temporal patterns.*
>
> *Specialised models are usually designed and pretrained for certain task (e.g., forecasting). They achieve strong performance on those tasks while not easily adapt to other task types (e.g., QA and imputation). Furthermore, traditional time series models often focus on periodic or regular signals. Their effectiveness on complex scientific time series with irregular dynamics or weak periodicity remains unclear.*
>
> *TimeOmni employs a lightweight time series encoder that equips LLMs with the ability to understand and generate time series while remaining compatible with general-purpose LLM training."*
>
> **[Regarding Inference Cost]**
>
> The patch–expert router is rule-based and therefore adds essentially no extra computational cost during inference. It performs deterministic routing without additional forward passes or parameters. To keep the output length to approximately $L$ regardless of the input signal length $T$, the router simply chooses the largest patch size $L_k$ such that $L_k < \frac{T}{L}$. This ensures stable token counts while scaling efficiently to long signals. The time-series encoder is correspondingly lightweight, containing only ~12 million parameters. As a result, both training and inference costs remain comparable to those of the underlying LLM backbone and are substantially lower than the cases of multimodal LLM architectures that rely on large, dedicated audio or vision encoders. This efficiency makes the approach well-suited for integrating scientific time-series processing into general-purpose LLM pipelines.
>
> *(The rest of your comments will be addressed in Response 2&3).*

---

> ### Author Response · Authors · 2025-11-19
> **Response to Reviewer P26w (2/3)**
>
> **[Regarding Small-Scale Adaptation]**
>
> We finetuned two open-source models, one image-text LLM and one time-series foundation model, on a number of benchmark tasks due to limited time and resources. Fine-tuning a text-only LLM becomes impractical because time-series signals translate into very long sequences, making both tokenisation and training computationally expensive. Results are shown below, where SR, MAE, and MAPE stand for success rate, mean absolute error and mean absolute percentage error, separately.
>
>
> | Task  | Metric | Qwen2.5-VL-7B (zero-shot) | Qwen2.5-VL-7B (finetune) | TimeMoE-base (zero-shot) | TimeMoE-base (finetune) | TimeOmni |
> |-------|---------|----------------------------|----------------------------|---------------------------|---------------------------|-----------|
> | **ASU01** | Acc $\uparrow$ | 48.4 | 49.9 | – | – | 69.0 |
> |       | F1 $\uparrow$ | 2.9 | 8.6 | – | – | 73.5 |
> | **ASG02** | MAPE $\downarrow$  | 6.7 | 5.2 | – | – | 2.8 |
> |       | SR $\uparrow$ | 1.5 | 4.5 | – | – | 100.0 |
> | **ASU03** | Acc $\uparrow$ | 27.5 | 30.1 | – | – | 87.8 |
> |       | F1 $\uparrow$ | 11.3 | 13.5 | – | – | 72.8 |
> | **EAU01** | Acc $\uparrow$ | 75.8 | 75.8 | – | – | 80.9 |
> |       | F1 $\uparrow$ | 80.6 | 80.6 | – | – | 82.5 |
> | **EAG02** | MAPE $\downarrow$ | 53.5 | 54.3 | – | – | 2.2 |
> |       | SR $\uparrow$ | 99.5 | 99.7 | – | – | 100.0 |
> | **BIU03** | Acc $\uparrow$ | 16.7 | 15.9 | – | – | 89.6 |
> |       | F1 $\uparrow$ | 4.8 | 8.0 | – | – | 89.2 |
> | **ENG03** | MAE $\downarrow$  | - | 579.9 | 469.0 | 417.6 | 300.8 |
> |        | MAPE $\downarrow$ | - | 13.3 | 11.3 | 10.5 | 7.4 |
> |        | SR $\uparrow$ | 0 | 46.0 | 100.0 | 100.0 | 100.0 |
>
> 1. We first finetune a vision-language LLM, Qwen2.5VL-7B, with LoRA. Results show that finetuning yields performance improvement on some tasks, but the gains are limited possibly because representing time series as images compromises numerical precision. For the ENG03 task, where the model fails to follow instructions in the zero-shot setting, finetuning improves instruction-following ability, although more than half of the samples still fail to produce the required length.
> 2. We then fully finetune TimeMoE, a time series model specialised for forecasting. Although finetuning improves its performance on forecasting tasks, finetuning does not enable it to handle tasks it could not previously address, such as imputation or QA.
>
> We would like to highlight that the key insight from benchmarking on SciTS is the architectural limitations of existing models—namely, their inability to support the diverse signals and task types in SciTS. As shown in Fig. 4, the success rates and failure analyses demonstrate that these are structural issues, which cannot be resolved merely through finetuning (as further confirmed by the results above). TimeOmni should therefore be understood as **a proof-of-concept architecture** rather than a fully optimised, all-encompassing model. It serves as an illustrative step toward enabling future LLMs to robustly process and reason over diverse, complex scientific time series.
>
> *(The rest of your comments will be addressed in Response 3).*

---

> ### Author Response · Authors · 2025-11-19
> **Response to Reviewer P26w (3/3)**
>
> **[Regarding Ablation Studies]**
>
> We provide ablation results below to quantify the contribution of each component. Again, we report the results on the same set of benchmark tasks due to limited time and resources.
>
> | Task   | Metric | TimeOmni | TimeOmni w/o Reprogramming | TimeOmni w/o Router |
> |--------|--------|----------|--------------------------------------|------------------------------------|
> | **ASU01** | Acc $ \uparrow $ | 69.0  | 62.3  | 58.4 |
> |      | F1 $ \uparrow $  | 73.5  | 70.6 | 50.4 |
> | **ASG02** | MAPE $ \downarrow $ | 2.8   | 10.2 | 14.3 |
> | **ASU03** | Acc $ \uparrow $ | 87.8  | 86.8 | 56.5 |
> |      | F1 $ \uparrow $ | 72.8  | 77.0 | 25 |
> | **EAU01** | Acc $ \uparrow $ | 80.9  | 77.4 | 81.5 |
> |      | F1 $ \uparrow $  | 82.5  | 81.5  | 84.1 |
> | **EAG02** | MAPE $ \downarrow $ | 2.2   | 2.2   | 10.7 |
> | **BIU03** | Acc $ \uparrow $ | 89.6  | 80.5 | 73.5 |
> |      | F1 $ \uparrow $ | 89.2  | 80.7 | 79.4 |
> | **ENG03** | MAPE $ \downarrow $ | 7.4   | 7.4  | 7.5 |
>
> 1. **Ablation on the reprogramming module:** We replace the reprogramming module with a two-layer MLP using GELU activations and Dropout. This simplification leads to consistent performance degradation across most tasks. The results indicate that the reprogramming module provides stronger alignment between time-series inputs and the text LLM, improving robustness and generalization across diverse tasks.
> 2. **Ablation on the patch-expert routing module:** We replace the patch-expert routing mechanism with a fixed patch size of 1024 while keeping all other settings unchanged. Because SciTS contains very long signals, using a patch size smaller than 1024 would cause GPU out-of-memory, making 1024 the smallest feasible choice. This fixed patch size has little impact on medium-length tasks (e.g., EAU01), and can even slightly improve them, since using a single expert increases the effective training data for that patch. However, performance degrades on very long sequences (e.g., BIU03), where the model fails to capture long-range dependencies. Additionally, SciTS includes short sequences (e.g., ASU03) whose lengths are far below 1024. For these tasks, forcing a 1024-length patch collapses the entire sequence into a single patch, introducing structural inefficiencies that further hurt performance. Overall, this ablation demonstrates that patch-expert routing is essential for handling the wide range of temporal scales in scientific time series, from very short to extremely long.
>
>
>
>
> We thank the reviewer again for the comments. We would like to clarify that this paper is submitted to the *"Datasets and Benchmarks"* track, with the SciTS benchmark as its intended primary contribution: the first comprehensive benchmark for scientific time series generation and understanding. We sincerely appreciate the reviewers’ recognition of this contribution. Our extensive evaluation reveals that existing architectures fail to support all signal and task types in SciTS. We therefore present TimeOmni as a conceptual demonstration—not as a competing method, but as an illustrative step toward LLMs capable of handling complex and heterogeneous scientific signals. We hope our responses fully resolve your concerns.

---

### Author Response · Authors · 2025-12-01
**Response summary**

We thank all reviewers again for their thoughtful comments. We appreciate that all reviewers recognise the contribution of SciTS benchmark, including its diversity, significance, and extensive evaluation. SciTS is **the first comprehensive benchmark for scientific time series generation and understanding**, spanning 50k+ instances, 7 task types, and 43 domain-specific tasks across 12 scientific disciplines. Below, we summarise our responses to the main concerns raised by the reviewers.



**1. Regarding Evaluation and Positioning of TimeOmni:**

 -  **Purpose of TimeOmni** *(General)*. SciTS reveals fundamental architectural limitations of current MLLMs when facing heterogeneous scientific time series. We highlight that our primary objective is to surface this research question and stimulate further research, and TimeOmni is offered as an illustrative prototype, not a definitive solution, to demonstrate the feasibility of building a model that can engage with the SciTS benchmark.

 - **Zero-shot / out-of-domain evaluation** *(Reviewers pjb9,5o2d)*. We emphasise that the training data of TimeOmni does not (and realistically cannot) cover all SciTS domains and tasks. Consequently, several tasks (e.g., ENG04–05, URG02–04) are evaluated in the zero-shot manner, enabling assessment of the model’s generalisation to unseen tasks, as detailed in Appendix G.

 - **Baseline finetuning / small-scale adaptation** *(Reviewers P26w,pjb9,5o2d)*. To further address reviewers' concerns about fair comparison, we additionally finetuned a representative image–text LLM and a time-series foundation model. Results confirm that their limitations are inherent to their architectures and cannot be overcome through finetuning alone.

**2. Regarding TimeOmni Ablation Studies**

 -  In response to requests from *Reviewers P26w and 5o2d*, we conducted ablations on the reprogramming module and patch-expert routing mechanism. The results show that both components are essential for handling the broad range of temporal scales present in scientific time series.

**3. Clarifications and Explanations**

 - **TimeOmni design clarifications.** We provide expanded clarifications on TimeOmni design, including stability considerations *(Reviewer 5o2d)*, inference cost *(Reviewer P26w)*, normalisation approaches *(Reviewer 5o2d)*, extensions beyond TimeLLM framework *(Reviewer 4z7J)*, and its support for multivariate and dynamic-length inputs *(Reviewers pjb9, 5o2d)*.

 - **Pointing to existing content.** For questions  already addressed in the paper, we point out the relevant sections, including detailed recent work *(in Appendix B.1, by Reviewer P26w)*, full results on per-task performance *(in  Appendix G, by Reviewer pjb9)*, metric computation under task failure *(in Section 5 (line 296-301), by Reviewer pjb9)*, and data provenance *(in Appendix D.2, by Reviewer 5o2d)*.

We note that some concerns arise from expectations of a full methodological contribution. We emphasise that this submission is to the **Datasets and Benchmarks track**, where SciTS is the primary intended contribution. SciTS opens an important and previously underexplored research direction, and TimeOmni is presented as an initial prototype. The introduction of TimeOmni is solely to demonstrate the feasibility of modelling and benchmarking SciTS tasks, and to encourage the community to pursue stronger future solutions. Enabling LLMs to effectively process such signals is crucial for building artificial general intelligence systems capable of supporting scientific analysis and discovery. We address each reviewer’s comments in detail below their respective review. We believe the revisions and clarifications provided adequately address all concerns.

---

### Meta-Review · Area_Chair_b5Dw · 2026-01-02

**Summary:**

This submission introduces SciTS, a large-scale and carefully curated benchmark for scientific time series understanding and generation across diverse domains and task types, along with TimeOmni as an illustrative framework to demonstrate feasibility. The paper is clearly written, well-structured, and addresses an important and under explored problem at the intersection of LLMs and scientific time series. The evaluation is broad and systematic, covering multiple model families and revealing structural limitations of existing approaches.

Overall, reviewers largely agree that the benchmark itself constitutes a significant contribution. While opinions diverged on the methodological novelty of TimeOmni, the majority view is that the benchmark and empirical findings provide substantial value to the community.

**Reviewer Concerns:**

Most substantive concerns raised by reviewers (regarding fairness of comparisons, lack of ablations, zero-shot vs. fine-tuned evaluation, architectural clarity, and per-task analyses) were adequately addressed in the rebuttal. The authors provided additional ablation studies, clarified training and evaluation protocols (including zero-shot and out-of-domain settings), and added analyses demonstrating that the observed limitations of baseline models are architectural rather than data-related. These responses substantially strengthened the paper and resolved the main technical questions raised by reviewers who initially expressed reservations.

One reviewer provided a low score with limited technical justification and did not meaningfully engage with the rebuttal (or subsequent clarifications). Other reviewers acknowledged that their primary concerns were addressed and expressed openness toward acceptance.

**Reviewer Scores:**

Two reviewers scored the paper 6, indicating acceptance-level quality, and one reviewer scored 4 (but based on the rebuttal and discussion, it is reasonable to believe that the reviewer with score 4 would likely have increased their score had post-rebuttal score updates been permitted).

The single score of 2 appears to be an outlier, both in tone and in depth of technical engagement, and is not well aligned with the broader reviewer consensus.

---

### Decision · Program_Chairs · 2026-01-26

Accept (Poster)